# Tropospheric ozone in CMIP6 Simulations

Paul T. Griffiths[1,2,*], Lee T. Murray[3,*], Guang Zeng[4], Youngsub Matthew Shin[1], N. Luke Abraham[1,2], Alexander T. Archibald[1,2], Makoto Deushi[5], Louisa K. Emmons[6], Ian E. Galbally[7,8], Birgit Hassler[9], Larry W. Horowitz[10], James Keeble[1,2], Jane Liu[11], Omid Moeini[12], Vaishali Naik[10], Fiona M. O'Connor[13], Naga Oshima[5], David Tarasick[12], Simone Tilmes[6], Steven T. Turnock[13], Oliver Wild[14], Paul J. Young[14,15], and Prodromos Zanis[16]

[1]Centre for Atmospheric Science, Cambridge University, UK
[2]National Centre for Atmospheric Science, Cambridge University, UK
[3]Department of Earth and Environmental Sciences, University of Rochester, Rochester, NY USA
[4]National Institute of Water and Atmospheric Research, Wellington, New Zealand
[5]Meteorological Research Institute, Tsukuba, Japan
[6]National Centre for Atmospheric Research, Boulder, Colorado
[7]Climate Science Centre, CSIRO Aspendale Victoria Australia
[8]Centre for Atmospheric Chemistry, University of Wollongong, Wollongong, NSW, Australia
[9]Deutsches Zentrum für Luft- und Raumfahrt (DLR), Institut für Physik der Atmosphäre, Oberpfaffenhofen, Germany
[10]NOAA Geophysical Fluid Dynamics Laboratory, Princeton, NJ, USA
[11]Department of Geography, University of Toronto, Canada
[12]Air Quality Research Division, Environment and Climate Change Canada
[13]Met Office Hadley Centre, Exeter, UK
[14]Lancaster Environment Centre, Lancaster University, Lancaster, UK
[15]Centre of Excellence for Environmental Data Science (CEEDS), Lancaster University, Lancaster, UK
[16]Department of Meteorology and Climatology, Aristotle University of Thessaloniki, Greece
[*]These authors contributed equally to this work.

**Correspondence:** Paul Griffiths (paul.griffiths@ncas.ac.uk); Lee T. Murray (lee.murray@rochester.edu)

**Abstract.** The evolution of tropospheric ozone from 1850 to 2100 has been studied using data from Phase 6 of the Coupled Model Intercomparison Project (CMIP6). We evaluate long-term changes using coupled atmosphere-ocean chemistry-climate models, focusing on the CMIP Historical and ScenarioMIP ssp370 experiments, for which detailed tropospheric ozone diagnostics were archived. The model ensemble has been evaluated against a suite of surface, sonde, and satellite observations of the past several decades, and found to reproduce well the salient spatial, seasonal and decadal variability and trends. The multimodel mean tropospheric ozone burden increases from $247 \pm 36$ Tg in 1850 to a mean value of $356 \pm 31$ Tg for the period 2005-2014, an increase of 44 %. Modelled present-day values agree well with previous determinations (ACCENT: $336 \pm 27$ Tg; ACCMIP: $337 \pm 23$ Tg and TOAR: $340 \pm 34$ Tg). In the ssp370 experiments, the ozone burden increases to $416 \pm 35$ Tg by 2100. The ozone budget has been examined over the same period using lumped ozone production ($P_{O3}$) and loss ($L_{O3}$) diagnostics. Both ozone production and chemical loss terms increase steadily over the period 1850 to 2100, with net chemical production ($P_{O3}$-$L_{O3}$) reaching a maximum around the year 2000. The residual term, which contains contributions from stratosphere-troposphere transport reaches a minimum around the same time, before recovering in the 21st century, while

dry deposition increases steadily over the period 1850-2100. Differences between the model residual terms are explained in terms of variation in tropopause height and stratospheric ozone burden.

## 1 Introduction

Tropospheric ozone ($O_3$) is an important component of air pollution and an oxidising species with adverse effects on human health (Jerrett et al., 2009; Turner et al., 2015; Malley et al., 2017) and vegetation (Fowler et al., 2009). It is also a greenhouse gas (GHG) with a radiative forcing of $0.4 \pm 0.2$ $Wm^{-2}$ (Stevenson et al., 2013; Myhre et al., 2013) and plays an important role in controlling the strength of the terrestrial carbon sink (Sitch et al., 2007). Ozone is not emitted directly into the troposphere but is produced there by the photochemical oxidation of carbon monoxide (CO), methane ($CH_4$) and non-methane volatile organic compounds (NMVOCs) in the presence of nitric oxide (NO) and nitrogen dioxide ($NO_2$). The tropospheric ozone burden is controlled by the balance between chemical production and loss processes, deposition at the surface and downward transport from the stratosphere.

In addition to its roles as a GHG and air pollutant, ozone is an oxidant and a precursor for the hydroxyl (OH) radical. OH (and by implication ozone) controls the lifetime of methane (Voulgarakis et al., 2013), the second most important anthropogenic GHG after carbon dioxide (Myhre et al., 2013). Oxidant levels mediate the formation of secondary aerosols such as sulfate and nitrate and play a major role in the aerosol budget and burden with important consequences for radiative forcing (Shindell et al., 2009; Karset et al., 2018). Accurate knowledge of ozone and how ozone has evolved since pre-industrial times is therefore critical to our understanding of the radiative forcing from aerosol and greenhouse gases (GHGs).

The lifetime of ozone in the troposphere varies considerably with location and season, ranging from a few hours in polluted urban regions up to a few weeks in the upper troposphere (Monks et al., 2015) and the global mean tropospheric lifetime is estimated to be 23.4±2.2 days (Young et al., 2013). Ozone has a sufficiently long lifetime in the troposphere to be transported over long distances, and this transport may therefore be affected by climate variability and by the associated changes in large-scale atmospheric circulation patterns that occur on interannual to decadal time scales. Emissions of ozone precursors from natural sources (e.g., lightning, vegetation, fires) also respond to natural variability contributing to large scale variability in ozone.

Due to the difficulties of measuring tropospheric ozone on a global scale, the global burden and budget are estimated using global atmospheric chemistry models which include chemistry climate models (CCMs), chemistry transport models (CTMs) and chemistry general circulation models (chemistry GCMs) (Young et al., 2018). While the tropospheric ozone burden and distribution during pre-industrial times is unknown from observations (Tarasick et al., 2019), the present-day ozone monitoring network can be used to calculate the tropospheric ozone burden and evaluate global atmospheric chemistry models. Multiple satellite products from Infrared Atmospheric Sounding Interferometer (IASI) such as IASI-FORLI and IASI-SOFRID corroborated by the Trajectory-mapped Ozonesonde dataset for the Stratosphere and Troposphere (TOST) indicate an overall mean present-day (2010-2014) global tropospheric ozone burden of 338±6 Tg in broad agreement with the current range of model estimates (Gaudel et al., 2018).

Recently, Young et al. (2018) presented an updated regional evaluation of tropospheric ozone simulated by models contributing to the Atmospheric Chemistry and Climate Model Intercomparison Project (ACCMIP) using data from: ozonesonde measurements, a new compilation of long-term measurements conducted aboard commercial aircraft of internationally operating airlines (MOZAIC-IAGOS), and a comprehensive database of global surface ozone measurements that was compiled within the Tropospheric Ozone Assessment Report (TOAR) framework. This evaluation revealed that the models are biased high in the Northern Hemisphere (NH) and low in the Southern Hemisphere (SH), with the biases generally persisting throughout the depth of the troposphere in agreement with previous global model evaluation studies (Fiore et al., 2012; Stevenson et al., 2013). Most CCMs capture the seasonal cycle of surface and free tropospheric ozone over most regions reasonably well, giving confidence in the relative contribution of the seasonal cycle of emissions and meteorology to the simulated seasonal cycle in ozone. However, there are still model deficiencies in simulating the seasonality of free tropospheric ozone in regions such as Equatorial America, Japan and northern high latitudes (Young et al., 2018) and of near-surface ozone over northern and north-eastern Europe (Katragkou et al., 2015), reflecting poor simulation of local and regional dynamics or missing chemical processes, complicated by the uncertainty in ozone precursor emissions. The spatial patterns in annual mean surface ozone and regional features of free tropospheric ozone are generally captured by current global chemistry models (Tilmes et al., 2016; Hu et al., 2017) including the ozone maximum west of southern Africa over the South Atlantic Ocean (Sauvage et al., 2007), the mid-Pacific minimum (Ziemke et al., 2010), and the summertime free tropospheric ozone maximum over the Eastern Mediterranean (Akritidis et al., 2016; Zanis et al., 2014).

The main chemical reactions contributing to tropospheric ozone production are reactions between NO and hydroperoxyl ($HO_2$) and other peroxyl radicals that are intermediate products of VOC degradation. Ozone chemical production occurs throughout the troposphere, particularly near the surface close to emissions, and also in the upper troposphere via lightning-produced $NO_x$. Deposition of ozone occurs at the surface via reactive chemical loss to surfaces. In the free troposphere, ozone loss by photolysis to produce $O^1D$, and the subsequent reaction of $O^1D$ with $H_2O$, and by chemical destruction involving reaction with hydroxy and hydroperoxyl radicals are important (Ayers et al., 1992).

The ozone source and sink terms vary between models due to differing approaches in representing the processes involved, and also due to differences in how these budget terms are defined (Stevenson et al., 2006; Young et al., 2013, 2018). Key issues include the representation of NMVOC chemistry which affects chemical production and loss terms, surface loss processes, and stratospheric influences. The definition of the tropopause will also influence the diagnosed burden and any influx from the stratosphere. The Tropospheric Ozone Assessment Report reviewed the ozone budget terms using results from models that took part in ACCENT and ACCMIP model intercomparisons and from recent single model studies (Young et al., 2018). They reported budget terms for the nominal year 2000, calculating a multi-ensemble mean global tropospheric ozone burden of 340 $\pm$ 34 Tg, chemical production of 4937 $\pm$ 656 Tg $O_3$ per year, chemical loss of 4442 $\pm$ 570 Tg per year, and deposition loss of 996 $\pm$ 203 Tg per year, leaving a residual term of 535 $\pm$ 161 Tg /year, which is assumed to represent the net stratospheric influx (Archibald et al., 2020a).

During the 21$^{st}$ century, changes in climate, stratospheric ozone-depleting substances (ODSs) and emissions of ozone precursor species are expected to be the major factors governing the amount of ozone and its distribution in the stratosphere,

the free troposphere and at the surface (Fiore et al., 2015; Revell et al., 2015). Changes in ozone precursor emissions have the largest effect on future tropospheric ozone concentrations, and precursor emission scenarios described by shared socioeconomic pathways (SSPs) and representative concentration pathways (RCPs) show reductions that would drive a decrease in ozone. A strong sensitivity to emission scenarios is supported by previous and recent model results that reveal a net decrease in the global tropospheric burden of ozone in 2100 compared to that in 2000 for all RCPs except RCP8.5, which shows an increase due to much larger methane concentrations than the other pathways (Stevenson et al., 2006; Naik et al., 2013; Banerjee et al., 2016; Sekiya and Sudo, 2014; Meul et al., 2018; Revell et al., 2015; Young et al., 2013).

The future evolution of methane concentrations and the emission of ozone precursors, such as biogenic volatile organic compounds (BVOCs), are a major source of uncertainty among the scenarios but there are also other sources of uncertainty related to GHG-induced climate change. Future changes in the net influx of ozone from the stratosphere to the troposphere are linked to changes in the stratospheric Brewer-Dobson circulation (BDC) and the amount of ozone in the lowermost stratosphere which are strongly influenced in a changing climate by changes in ODSs and long-lived GHGs. Future decreases in ODSs will lead to an ozone increase throughout the atmosphere with the largest percentage changes in the upper stratosphere and in the high-latitude lower stratosphere (with a particularly large impact on the SH). However, changes in GHGs will lead to a more complex pattern of ozone changes, with increases of ozone in the upper stratosphere (from GHG-induced cooling slowing the rate of gas phase ozone loss) and an increase of net stratospheric influx due to a possible strengthening of the BDC, with ODS decreases counteracting such a strengthening of the BDC due to GHG increases (Morgenstern et al., 2018; Polvani et al., 2018, 2019). Lu et al. report that the increases in tropospheric ozone in the SH are the result of circulation changes (Lu et al., 2019). For the coming decades, future net changes in the BDC depend on the climate change scenario and compliance with the Montreal Protocol. The BDC acceleration in response to increased GHG forcing is a robust finding across a range of atmospheric models with varying representations of the stratosphere (Butchart, 2014; Oberländer-Hayn et al., 2016) although there are still uncertainties in the magnitude (Morgenstern et al., 2018) and attribution of the strengthening. The substantial weakening effect of ODS decreases on the BDC has only recently been established (Morgenstern et al., 2018; Polvani et al., 2018, 2019). Banerjee et al. (2016) reported that a strengthened BDC under the RCP8.5 scenario has the strongest effect on tropospheric ozone in the tropics and subtropics, while stratospheric ozone recovery from declining long-lived ODSs has a larger role in the mid-latitudes and extratropics. Meul et al. (2018) suggested that the global annual mean influx of stratospheric ozone into the troposphere will increase by 53 % between the years 2000 and 2100 under the RCP8.5 greenhouse gas scenario and that this will be smaller for the moderate RCP6.0 scenario, but the relative change in the contribution of ozone of stratospheric origin in the troposphere is of comparable magnitude in both scenarios.

While all studies agree that changes to net stratosphere to troposphere transport will tend to increase future tropospheric ozone, the relative importance of stratosphere-troposphere exchange (STE) of ozone versus in-situ net chemical production for future tropospheric ozone trends remains uncertain. A study using new simulations from multiple CCMs finds considerable disagreement among models regarding past and future responses to drivers of tropospheric ozone even when the same scenario is considered, with much of the model spread likely due to the uncertainty in impacts on ozone in the tropopause region driving inter-model variations in STE trends (Morgenstern et al., 2018). In addition to these stratospheric influences, further

uncertainty arises from inter-model differences in tropospheric chemistry and physics (such as photolysis, convection and the boundary-layer scheme).

In this study, we examine the evolution of tropospheric ozone and describe the changes to the budget using the common model diagnostics of ozone production, loss and dry deposition to the surface. Our study focuses on transient simulations that were performed for CMIP6. The simulations run from preindustrial times to the present-day (i.e., the CMIP "Historical" simulations of the CMIP6) and from the present-day to end of the 21$^{st}$ century (i.e. "ssp370" of the future ScenarioMIP simulations) (Eyring et al., 2016). Five models including interactive stratospheric chemistry are selected for this analysis, which differs from previous multi-model studies (e.g., Stevenson et al., 2006; Young et al., 2013). CMIP6 builds on the approach of the Chemistry Climate Model Intercomparison (CCMI) project using long transient simulations but adds more diagnostics and a new, more complete set of emission data, and the most up-to-date and complete/complex set of interactive models. It draws on an improved set of observational constraints via TOAR to provide a comprehensive set of evaluation of the models' performance against well-established metrics (section 3) for recent decades, and evolution of the tropospheric ozone burden and budget over the full period of the experiments of 1850 to 2100 (section 4).

This paper forms part of a set of papers in support of Intergovernmental Panel on Climate Change (IPCC) Sixth Assessment Report. Other papers published or under discussion at the time of writing feature an analysis of chemistry and feedbacks (Thornhill et al., 2020), stratospheric ozone (Keeble et al., 2020), ozone radiative forcing (Skeie et al., 2020; Morgenstern et al., 2020), air pollution and particulate matter (Turnock et al., 2020; Allen et al., 2020), and oxidising capacity (Stevenson et al., 2020).

## 2    Models, Simulations and Configuration Details

### 2.0.1    GFDL-ESM4

The atmospheric component of the GFDL-ESM4 (Dunne et al., 2019) called AM4.1, includes an interactive tropospheric and stratospheric gas-phase and aerosol chemistry scheme (Horowitz et al., 2020). The model includes 56 prognostic (transported) tracers and 36 diagnostic (non-transported) chemical species, with 43 photolysis reactions, 190 gas-phase kinetic reactions, and 15 heterogeneous reactions. The tropospheric chemistry includes reactions for the $NO_x$-$HO_x$-Ox-CO-$CH_4$ system and oxidation schemes for other NMVOCs. The stratospheric chemistry accounts for the major ozone loss cycles (Ox, $HO_x$, $NO_x$, $ClO_x$, and $BrO_x$) and heterogeneous reactions on liquid and solid stratospheric aerosols as in Austin et al. (2012). The chemical system is solved using an implicit Euler backward method with Newton-Raphson iteration. Photolysis rates are calculated interactively using the FAST-JX version 7.1 code, accounting for the radiative effects of simulated aerosols and clouds. Emissions of BVOCs, including isoprene and monoterpenes, are calculated online in AM4.1 using the Model of Emissions of Gases and Aerosols from Nature (MEGAN; (Guenther et al., 2006)), as a function of simulated air temperature and shortwave radiative fluxes. Details on the chemical mechanism are included in Horowitz et al. (2020). The gas-phase and heterogeneous chemistry configuration is similar to that used by Schnell et al. (2018). Anthropogenic and biomass burning

emissions are prescribed from the dataset of Hoesly et al. (2018) and van Marle et al. (2017a) developed in support of CMIP6. Natural emissions of ozone precursors not calculated interactively are prescribed in the same way as in Naik et al. (2013).

The bulk aerosol scheme, including 18 transported aerosol tracers, is similar to that in AM4.0 (Zhao et al., 2018), with the following updates: (1) ammonium and nitrate aerosols are treated explicitly, with ISORROPIA (Fountoukis and Nenes, 2007) used to simulate the sulfate–nitrate–ammonia thermodynamic equilibrium; (2) oxidation of sulfur dioxide and dimethyl sulfide to produce sulfate aerosol is driven by the gas-phase oxidant concentrations (OH, $H_2O_2$, and ozone) and cloud pH simulated by the online chemistry scheme, and (3) the rate of aging of black and organic carbon aerosols from hydrophobic
to hydrophilic forms varies with calculated concentrations of hydroxyl radical (OH). Sources of secondary organic aerosols (SOA) include an anthropogenic source from oxidation of the simulated $C_4H_{10}$ hydrocarbon tracer by hydroxyl radical and a biogenic pseudo-emission scaled to BVOC emissions from vegetation.

### 2.0.2   UKESM1-LL-0

UKESM1-LL-0 (also abbreviated to "UKESM1" here) is the UK's Earth System Model (Sellar et al., 2019). It is based on the
Global Coupled 3.1 (GC3.1) configuration of HadGEM3 (Williams et al., 2018), to which various Earth system components have been added e.g. ocean biogeochemistry, terrestrial carbon/nitrogen cycle, and atmospheric chemistry. The atmospheric and land components are described in Walters et al. (2019). The chemistry scheme included in UKESM1 is a combined stratosphere-troposphere chemistry scheme (Archibald et al., 2020b) from the UK Chemistry and Aerosol (UKCA) model, combining the stratospheric chemistry scheme of Morgenstern et al. (2009) with the tropospheric (TropIsop) chemistry scheme
of O'Connor et al. (2014). A paper describing and evaluating this stratosphere-troposphere scheme in UKESM1 is currently in discussion (Archibald et al., 2020b). The aerosol scheme is a two-moment scheme from UKCA, called GLOMAP-mode, and is part of the Global Atmosphere 7.0/7.1 configuration of HadGEM3 (Walters et al., 2019). It models sulphate, sea salt, organic carbon and black carbon. Some improvements to the aerosol scheme for GA7.1 were required to address the strong negative aerosol forcing found with GA7.0 and are documented in Mulcahy et al. (2018). Dust is modelled separately in 6 size
bins following a variant of the Woodward scheme. Further discussion of the aerosol radiative forcing UKESM1 are given in (Mulcahy et al., 2020).

Anthropogenic and biomass burning emissions are prescribed (Hoesly et al., 2018; van Marle et al., 2017a) but emissions of isoprene and monoterpenes are interactive, and are based on the interactive biogenic VOC (iBVOC) emission model (Pacifico et al., 2011). Lightning emissions of $NO_x$ ($LNO_x$) are also interactive using the cloud top height parameterization of Price
and Rind (Price and Rind, 1992, 1993). Other natural emissions are prescribed as climatologies and will be discussed fully in Archibald et al. (2020b). For volcanic eruptions, internally-consistent stratospheric Aerosol optical depth (AOD) and surface area density (SAD) are prescribed for both the volcanic forcing and for the UKCA stratospheric heterogeneous chemistry.

### 2.0.3   CESM2-WACCM

CESM2-WACCM uses the Community Earth System Model version 2, (Emmons et al., 2020), and is a fully coupled Earth
System Model. The Whole Atmosphere Community Climate Model version 6 (WACCM6) is coupled to the other components

in CESM2. The Parallel Ocean Program version 2 (POP2) (Smith and Gent, 2002; Danabasoglu et al., 2012) includes several improvements compared to earlier versions, including ocean biogeochemistry represented by the Marine Biogeochemistry Library (MARBL), which incorporates the Biogeochemical Elemental Cycle (BEC) ocean biogoechemistry-ecosystem model (e.g., Moore et al., 2013). Additional components are the sea-ice model CICE version 5.1.2 (CICE5) (Hunke et al., 2015) and the Community Ice Sheet Model version 2.1 (CISM2.1), (Lipscomb et al., 2019). The Community Land Model version 5 (CLM5) also includes various updates, including interactive crops and irrigation for the land , and the Model for Scale Adaptive River Transport (MOSART).

CESM2-WACCM has a good representation of the tropospheric dynamics and climate, and also simulates internal variability in the stratosphere, including Stratospheric Sudden Warming (SSW) events on the intraseasonal timescales and the explicitly-resolved Quasi-Biennial Oscillation (Gettelman et al., 2019). The CESM2-WACCM model includes interactive chemistry and aerosols for the troposphere, stratosphere and lower thermosphere with 228 chemical compounds, including the 4-mode Modal Aerosol Model (MAM4) (Emmons et al., 2020). In particular, it includes an extensive representation of secondary organic aerosols based on the Volatility Basis Set (VBS) model framework (Tilmes et al., 2019) following the approach by (Hodzic et al., 2016). The scheme includes both updates to the SOA formation and removal pathways. MAM4 has been further modified to incorporate a new prognostic stratospheric aerosol capability (Mills et al., 2016). The modifications include mode width changes, growth of sulfate aerosol into the coarse mode, and the evolution of stratospheric sulfate aerosols from natural and anthropogenic emissions of source gases, including carbonyl sulfide (OCS) and volcanic sulfur dioxide (SO2). Anthropogenic and biomass burning emissions are prescribed (Hoesly et al., 2018; van Marle et al., 2017a). Biogenic emissions including BVOC are produced from the Model of Emissions of Gases and Aerosols from Nature (MEGAN) version 2.1 (Guenther et al., 2012) and and are also used for SOA formation.

### 2.0.4 GISS-E2-1-G

GISS-E2-1-G is the NASA Goddard Institute for Space Studies (GISS) chemistry-climate model version E2.1 using the GISS Ocean v1 (G01) model. The model configurations submitted for CMIP6 are described in detail by Kelley et al. (2020) and Miller et al. (2020). Here, we use the subset of model configurations that ran with online interactive chemistry. The atmospheric component was run with horizontal resolution of 2° latitude by 2.5° longitude with 40 hybrid sigma-pressure vertical layers extended from the surface to 0.1 hPa ($\sim$28 in the troposphere). Online interactive chemistry follows the GISS Physical Understanding of Composition-Climate INteractions and Impacts (G-PUCCINI) mechanism for gas-phase chemistry (Shindell et al., 2001, 2003, 2006, 2013; Kelley et al., 2020) and either the One-Moment Aerosol (OMA) or the Multiconfiguration Aerosol TRacker of mIXing state (MATRIX) model for the condensed phase (Bauer et al., 2020). The gas-phase mechanism includes 146 reactions (including 28 photodissociation reactions) acting on 47 species throughout the troposphere and stratosphere including five heterogeneous reactions. The model advects 26 (OMA) or 51 (MATRIX) aerosol particle tracers and 34 gas-phase tracers. Anthropogenic and biomass burning emissions are prescribed following the CMIP6 guidelines. Lightning $NO_x$ emissions are calculated online in deep convection as described by Kelley et al. (2020). Soil microbial $NO_x$ emissions are prescribed from climatology. Biogenic emissions of isoprene are calculated online and respond to temperature (Shindell et al.,

2006), but are prescribed for alkenes, paraffins and terpenes. Methane is prescribed as a surface boundary condition but allowed to advect and react with the chemistry in the historical runs and a subset of the SSP simulations; some future simulations used interactive online methane emissions following Shindell et al. (2004). The atmosphere is coupled to the GISS Ocean v1 (GO1) model (Kelley et al., 2020) with a horizontal resolution of 1° latitude by 1.25° longitude with 40 vertical levels.

### 2.0.5 MRI-ESM2-0

MRI-ESM2-0 is the Meteorological Research Institute (MRI) Earth System Model (ESM) version 2.0. Detailed descriptions of the model and evaluations are given by (Yukimoto et al., 2019a; Kawai et al., 2019; Oshima et al., 2020). MRI-ESM2-0 consists of four major component models: an atmospheric general circulation model with land processes (MRI-AGCM3.5), an ocean–sea-ice general circulation model (MRI Community Ocean Model version 4, MRI.COMv4), an aerosol chemical transport model (Model of Aerosol Species in the Global Atmosphere mark-2 revision 4-climate, MASINGAR mk-2r4c), and an atmospheric chemistry model (MRI Chemistry Climate Model version 2.1, MRI-CCM2.1). A coupler is used to interactively couple each component model (Yoshimura and Yukimoto, 2008). MRI-ESM2-0 uses different horizontal resolutions in each atmospheric component model but employs the same vertical resolution, namely MRI-AGCM3.5, the aerosol model, and the atmospheric chemistry model use TL159 (approximately 120 km or 1.125° x 1.125°), TL95 (approximately 180 km or 1.875° x 1.875°), and T42 (approximately 280 km or 2.8125° x 2.8125°), respectively, and all models employ 80 vertical layers (from the surface to the model top at 0.01 hPa) in a hybrid sigma-pressure coordinate system. MRI.COMv4 uses a tripolar grid with a nominal horizontal resolution of 1° in longitude and 0.5° in latitude with 60 vertical layers (Tsujino et al., 2017). Detailed descriptions of the CMIP6 CMIP historical experiments by MRI-ESM2-0 are given by (Yukimoto et al., 2019b).

MRI-ESM2-0 includes interactive chemistry and aerosols in the atmosphere. The atmospheric chemistry model, MRI-CCM2.1, calculates evolution and distribution of the ozone and other trace gases in the troposphere and middle atmosphere (Yukimoto et al., 2019b) and (Deushi and Shibata, 2011) . The model includes 64 prognostic chemical species and 24 diagnostic chemical species, with 184 gas-phase reactions, 59 photolysis reactions, and 16 heterogeneous reactions. It considers Ox-HOx-NOx-CH$_4$-CO chemical system and NMVOC oxidation reactions, as well as the major stratospheric chemical system. Anthropogenic and biomass burning emissions are prescribed (Hoesly et al., 2018; van Marle et al., 2017b). Lightning emissions of NO$_x$ are diagnosed at 6-h intervals following the parameterization of Price and Rind (Price and Rind, 1992, 1993). Other natural emissions such as biogenic, soil, and ocean emissions are prescribed as climatologies (Deushi and Shibata, 2011). The aerosol component model, MASINGAR mk-2r4c, calculates the physical and chemical processes of the atmospheric aerosols and treats the following species: non-sea-salt sulfate, black carbon, organic carbon, sea salt, mineral dust, and aerosol precursor gases (Yukimoto et al., 2019b; Oshima et al., 2020). The size distributions of sea salt and mineral dust are divided into 10 discrete bins, and the sizes of the other aerosols are represented by lognormal size distributions.

## 2.1 Simulations

For this review, we used available data from the CMIP6 CMIP Historical experiments from UKESM1 (Tang et al., 2019) , GFDL-ESM4 (Krasting et al., 2018), GISS-E2-1-G (NASA Goddard Institute For Space Studies (NASA/GISS), 2019), MRI-

ESM2-0 (Yukimoto et al., 2019b) and CESM2-WACCM (Danabasoglu, 2019a). For ScenarioMIP ssp370 experiments we used data archived by UKESM1 (Good et al., 2019), GFDL-ESM4 (John et al., 2018), GISS-E2-1-G (NASA Goddard Institute For Space Studies (NASA/GISS), 2020), MRI-ESM2-0 (Yukimoto et al., 2019c) and CESM2-WACCM (Danabasoglu, 2019b).

We analysed those models that had archived sufficient data to the Earth System Grid Federation Peer-to-Peer system to permit accurate characterisation of the tropospheric ozone burden. In practice this meant, we used archived ozone data from the AERmon characterisation of the tropospheric ozone burden (variable name="o3") on native model grids, along with data on the tropopause pressure using the World Meteorological Organization (WMO) definition of the tropopause (variable name = "ptp"). For the budget calculations, dry deposition (variable name="dryo3"), chemical production (variable name="o3prod"), chemical destruction (variable name="o3loss") along with "airmass", air temperature(variable name="ta") and pressure diagnostics (variables such as "ps" and "phalf" where required) were used from the AERmon realm.

## 2.2 Emissions

Figure 1 shows the emissions and methane forcing used in the CMIP6 models. Data for the period 1850 to 2014 were taken from the CMIP6 CMIP Historical experiment, and for the period 2015 to 2100 from the ScenarioMIP ssp370 experiment.

CO emissions were calculated using the output "emico" variable output by each model. Anthropogenic $NO_x$ emissions used in each model were calculated as follows: for UKESM1-0-LL, the "eminox" variable was used which is the sum of anthropogenic, open-burning, soil, and aircraft $NO_x$ emissions; for GFDL-ESM4 and GISS-E2-1-G, the eminox variable represents anthropogenic, open-burning, soil, aircraft, and lightning $NO_x$ emissions, so the accompanying "emilnox" (lightning) output for these models were subtracted to calculate anthropogenic $NO_x$; finally, for CESM2-WACCM, the eminox variable consists of anthropogenic, open-burning, and soil $NO_x$ emissions, so a small fraction of total $NO_x$ emissions in the form of anthropogenic aircraft are missing. Biogenic non-methane volatile organic compound emissions were calculated using the "emibvoc" variable.

All five models used a version of the Price and Rind (1992) lightning flash parameterization that assumes lightning activity increases with increasing convective cloud height; since most models predict increases in convective depths with increasing greenhouse gas levels, this scheme generally predicts monotonic increases in lightning over time, from a multi-model mean of $4.9 \pm 1.9$ Tg(N) yr$^{-1}$ in 1850-1859 to $5.1 \pm 2.0$ Tg(N) yr$^{-1}$ in 2005-2014, and to $6.2 \pm 2.6$ Tg(N) yr$^{-1}$ by 2090-2099 (dashed lines of top left panel of Fig. 1). However, how lightning may respond to a warming world remains unknown (e.g., Williams, 2005; Price, 2013; Murray, 2016). Since lightning $NO_x$ has a disproportionately strong impact on tropospheric ozone burdens relative to surface emissions (e.g., Murray et al., 2013), this remains an important source of uncertainty both between models and in the temporal evolution of tropospheric ozone.

The CO and $NO_x$ tropospheric burdens were calculated by applying a tropospheric mask derived from each model's tropopause pressure/height output. The $NO_x$ burden was determined as the sum of the NO and $NO_2$ mole fraction outputs.

The prescribed methane lower boundary concentrations are described in Meinshausen et al. (2019). Over the ssp370 period, global methane concentrations increase monotonically.

## 3 Evaluation of tropospheric ozone over recent decades

Figure 2 shows the present-day spatial distribution of ozone and its inter-model variability in the CMIP6 ensemble. The spatial patterns are broadly consistent with observations (see Sects. 3.1-3.4) and those of earlier model intercomparison studies (e.g., Stevenson et al., 2006; Young et al., 2013). Zonal mean mixing ratios are highest in the upper troposphere, especially in the extratropics, reflecting longer chemical lifetimes at higher altitude (Fig. 2a). Ozone is also higher in the NH relative to the SH, reflecting higher rates of stratospheric downwelling (e.g., Rosenlof, 1995) and surface ozone precursor emissions. The model ensemble members are in relative good agreement, with a standard deviation of less than 25 % throughout most of the troposphere. There is improved multi-model agreement in the northern hemisphere and a slight degradation in the southern hemisphere relative to Young et al. (2013), although it is hard to assess given the different number of ensemble members between the two assessments (15 then vs. 5 here). The greatest absolute and relative differences in mixing ratio occur in the upper troposphere. This reflects relatively large inter-model variability in the simulated mean tropopause pressure ($\pm$ 30 hPa). The tropopause acts as a dynamical barrier that separates the high-ozone air of the stratosphere from the low-ozone air of the troposphere. Therefore, simulated differences in tropopause height manifest as large differences in ozone mixing ratio in the upper troposphere and lower stratosphere (UT/LS) region. Furthermore, variations in tropopause pressure allow for more or less air mass to exist in the troposphere ($\pm \sim$3 %), also contributing to variations in tropospheric columns of ozone (TCO) between models, especially in the northern extratropics (Fig. 2e-f). Inter-model variability in TCO (Fig. 2e) is about twice as high as earlier model intercomparison studies (e.g., Young et al., 2013) due to our use of the thermal tropopause rather than a chemical tropopause (see Sect. 3.4). Ozone also has relatively large inter-model variability in the southern extratropical free troposphere, likely resulting from the relatively large variability in southern lower stratospheric ozone and subsequent transport across the tropopause. In addition, ozone mixing ratios vary relatively largely between models in the tropics, especially in the surface boundary layer (especially in regions of high biogenic emissions such as the Amazon) and the UT/LS region. The latter is of interest due to the importance of absorption of outgoing longwave radiation for radiative forcing in this region (e.g., Forster and Shine, 1997).

### 3.1 Surface ozone

Figure 3 compares the CMIP6 model ensemble to five remote surface ozone stations with the longest available *in situ* sampling record: Mauna Loa, Hawai'i, USA (MLO, 19.5°N, 155.6°W, 3397 m.a.s.l., 1957-present), the South Pole (SPO, 90°S, 59°E, 2840 m.a.s.l., 1961-present), Barrow, Alaska, USA (BRW, 71.3°N, 156.6°W, 11 m.a.s.l., 1973-present), Cape Matatula, Tutuila, American Samoa (SMO, 14.2°S, 170.6°E, 42 m.a.s.l., 1975-present), and Cape Grim, Tasmania, Australia (CGO, 40.7°S, 144.7°E, 94 m.a.s.l., 1982-present). The figure provides the respective trends, temporal correlation, and mean normalized bias error for the model ensemble and observations. These measurements in remote background locations are useful constraints for evaluation of trends in the tropospheric ozone budget. Mauna Loa is especially useful for evaluating trends in tropospheric ozone. In addition to a long historical record, it is a remote mountain site that frequently samples free tropospheric air masses.

For a more thorough evaluation and examination of surface ozone in the CMIP6 simulations, including implications for surface air quality, we refer the reader to the CMIP6 surface ozone companion paper (Turnock et al., 2020).

For Mauna Loa, we use monthly average surface ozone measured using a Regener type potassium iodide (KI) automatic ozone analyser for 1957-1959 and a UV photometric analyser for 1974-2014. At Barrow and American Samoa, surface ozone was measured using a UV photometric analyser for 1973-2014 and 1975-2014 respectively. At the South Pole, ozone was measured using a Regener type potassium iodide (KI) automatic ozone analyser for 1961-1963, a corrected Regener type chemiluminescent automatic ozone analyser for 1964-1966, an Electrochemical Cell analyser for 1967-1973 and a UV photo-

metric analyser for 1975-2014. Data for these four stations are archived at ftp://aftp.cmdl.noaa.gov/data/ozwv/SurfaceOzone/. Cape Grim surface ozone was measured using a UV photometric analyser for 1982-2014 and are available as hourly averages from the WMO World Data Centre for Reactive Gases at https://www.gaw-wdcrg.org. Monthly observations were converted to annual averages for those with 9 months or more of data. Corrections to the data to account for the different ozone analysers operated during the historical period have been applied to the SPO data using the framework described by Tarasick et al.

(2019). We sample the models at the surface level for Barrow, American Samoa, Cape Grim and the South Pole, and at the 680 hPa level for Mauna Loa.

    The models overestimate surface ozone concentrations at the two NH sites by 2-3 ppbv and the tropical SH site by 6 ppbv while underestimating surface ozone at the two extratropical SH sites by 1-7 ppbv. In particular, the models significantly underestimate surface ozone at the South Pole. In the time before and after polar sunrise at Barrow there are significant ozone-

330 depletion events in surface air that are large enough to affect annual mean ozone levels (e.g., Oltmans and Levy, 1994; Helmig et al., 2007) and perhaps suggest one reason for the model-observation difference. These discrepancies may also reflect biases associated with comparing point data to a much coarser model grid cell.

    At Barrow, Mauna Loa, American Samoa and Cape Grim, observed surface ozone has increased on average by 0.5-2.0 ppbv per decade (2-4 % per decade) since measurements began. Despite the mean bias, the models well-capture the magnitude of the decadal

trends in response to climate and emission forcings. In the Southern Hemisphere part of the trend in tropospheric ozone can be explained by the poleward expansion of the Hadley circulation (Lu et al., 2019). Over Antarctica, observations show an initial decrease from the 1960s through the mid-1990s, before ozone began rising, resulting in no significant trend during this period. The models underestimate the magnitude of the observed reduction, and consequently, simulate a small growth here.

## 3.2   Vertical, meridional and seasonal ozone distribution

Figure 4 compares the vertical, meridional and seasonal distribution of ozone in the CMIP6 ensemble to climatological measurements from ozonesondes (balloons). We use sonde measurements archived by the World Ozone and Ultraviolet Radiation Data Centre (WOUDC) of the World Meteorological Organization/Global Atmosphere Watch Program (WMO/GAW). The data were accessed on Nov. 4, 2019 from https://doi.org/10.14287/10000008. A total of 23,392 profiles using Carbon-Iodine (Komhyr, 1969), ECC (Komhyr, 1971), and Brewer-Mast (Brewer and Milford, 1960) sondes from 82 sites world-wide were

aggregated over the period 2005-2014. Sondes show a modest high bias in the troposphere of about 1-5 % $\pm$ 5 % when compared to more accurate UV-absorption measurements (Tarasick et al., 2019). Measurement precision is $\pm$3-5 % and the overall

uncertainty in ozone concentration is less than 10 % in the troposphere (Kerr et al., 1994; Smit et al., 2007; Tarasick et al., 2016, 2019).

The models reproduce the increase in ozone with altitude and from south to north, and well reproduce the seasonal cycle of ozone in the tropics and northern extratropics ($r^2$ all greater than 0.72). Note the northern hemispheric overestimate and southern hemispheric underestimate seen at the surface (Sect. 3.1) extends into the lower free troposphere. The ensemble mean is biased high by about 10 % in the NH, although always falls within the range of interannual variability in the observations (vertical lines). The ensemble reproduces the magnitude and seasonality of the southern tropics better than the other regions, although it fails to reproduce the timing and magnitude of the October peak associated with the zonal wave-one South Atlantic ozone maximum (Fishman et al., 1990, 1991; Shiotani, 1992; Thompson and Hudson, 1999; Thompson et al., 2000; Thompson, 2003b; Sauvage et al., 2006). The model ensemble performs worst in the southern extratropics, resulting from seasonal behavior anti-correlated with the observations in one model (GISS-E2-1-G); when that model is removed from the ensemble, the seasonal correlation at 500 hPa becomes $r = 0.96$ but the mean bias increases in magnitude to -3 %. CMIP6 shows nominal improvements in certain regions such as the southern tropics with respect to biases and correlations reported by the earlier ACCMIP (Young et al., 2013) and Atmospheric Composition Change: the European Network of excellence (ACCENT) (Stevenson et al., 2006) studies, although it is difficult to evaluate given the smaller number of models in the CMIP6 (5) versus ACCMIP (15) and ACCENT (26) studies, and given different periods of evaluation.

### 3.3 Tropospheric ozone column abundance

Satellites provide high-frequency near-global coverage of tropospheric columns of ozone (TCO), the amount of ozone integrated from the surface to the tropopause, typically given in Dobson Units (1 DU $\equiv 2.69 \times 10^{20}$ molecules m$^{-2}$). Figure 5 compares the seasonality of TCO in the model ensemble to that of the Ozone Monitoring Instrument/Microwave Limb Sounder (OMI/MLS) product (Ziemke et al., 2006). The OMI/MLS product is the residual of the OMI total ozone column and the MLS stratospheric ozone column, available as gridded $1° \times 1.25°$ monthly means, and is provided from 60°S to 60°N due to its reliance on solar backscattered UV radiation. Here we use the data for 2005-2014 downloaded in Nov. 2019 from https://acd-ext.gsfc.nasa.gov/Data_services/cloud_slice/new_data.html.

The model ensemble captures the salient features of spatial-seasonal patterns in TCO from OMI/MLS. This includes zonal-wide maxima in the subtropics (where isentropes intersect the tropopause), greater TCO in the NH, minima over the remote Pacific and Antarctic, and the zonal-wave pattern over the South Atlantic ocean. On average, the models overestimate TCO in the NH and Indian Ocean by up to 25 % versus OMI/MLS, and underestimate ozone in the remote Pacific and Southern Ocean, yielding small net positive biases when integrated over the whole region (+2 DU or 7-10 % in all seasons). The models show greatest disagreement in summertime extratropical TCO, especially in the high Arctic, but OMI/MLS is not available here.

Figure 6 evaluates annual mean TCO in the model ensemble versus OMI/MLS and the Trajectory-mapped Ozonesonde dataset for the Stratosphere and Troposphere (TOST). TOST is a global three-dimensional dataset of tropospheric and stratospheric ozone, derived from the ozonesonde record (Liu et al., 2013b, a). TOST determines TCO using 96-hour forward and backward trajectory calculations of the ozone profiles using the Hybrid Single-Particle Lagrangian Integrated Trajectory

(HYSPLIT) particle dispersion model (Draxler and Hess, 1997, 1998) driven by the global NOAA National Centers for Environmental Prediction/National Center for Atmospheric Research (NCEP/NCAR) pressure level meteorological reanalysis. By assuming ozone production and loss to be negligible, the ozone is mapped to other locations and times using a 3-dimensional grid of $5° \times 5° \times 1$ km. TCO is calculated from the surface to the tropopause, which is defined using the WMO 2 K/km lapse-rate definition applied to the NCEP reanalysis. Over mountainous areas a topographic correction is made in order to address an apparent bias in TCO over high mountains. TOST has been evaluated using individual ozonesondes, excluded from the mapping, by backward and forward trajectory comparisons, and by comparisons with aircraft profiles and surface monitoring data (Tarasick et al., 2010; Liu et al., 2013a, b). Differences are typically about 10 % or less, but there are larger biases in the UT/LS, the boundary layer, and in areas where ozonesonde measurements are very sparse. The accuracy of the TOST product depends largely on the accuracy of HYSPLIT and the meteorological data on which it is based.

The TOST data presented here uses the troposphere-only dataset, which explicitly excludes trajectories originating in the stratosphere. This avoids including stratospheric air, with its very high ozone content, when the NCEP tropopause is higher than the climatological tropopause (i.e. the ozone tropopause). If the same calculations are made using the full-profile TOST dataset, the calculated burden is on average 42 Tg (about 15 %) larger.

The models agree with the TOST product in much of the tropics, except in the remote Pacific, where they are biased low, qualitatively consistent with the OMI/MLS product. Since the TOST product is on average lower than OMI/MLS, especially in higher latitudes, the models are biased even higher with respect to the TOST data than OMI/MLS (+6 DU and 22 %).

### 3.4 Tropospheric ozone burden

Figure 7 compares the present-day tropospheric ozone burden to seven space-based satellite products and the ozonesonde-derived TOST product. The satellite-derived products include the annual mean burdens for 60°S-60°N from OMI/MLS, IASI (Infrared Atmospheric Sounding Interferometer)-FORLI (Fast Optimal Retrievals on Layers), IASI-SOFRID (SOftware for a Fast Retrieval of IASI Data), GOME (Global Ozone Monitoring Experiment)/OMI-SOA (Smithsonian Astrophysical Observatory), OMI-RAL (Rutherford Appleton Laboratory), SCIAMACHY (SCanning Imaging Absorption SpectroMeter for Atmospheric CHartographY), and TES (Tropospheric Emission Spectrometer) reported by Gaudel et al. (2018). The TOST record has been calculated since 1980, but is most accurate beginning in 1998 when sonde measurements began in the tropics as part of the Southern Hemisphere Additional OZonesondes (SHADOZ) campaign (Thompson, 2003a). The satellite burdens span a range of values ($\sim$250-350 Tg) consistent with the multi-model mean (MMM) and standard deviation, reflecting uncertainties in the tropopause definition (Gaudel et al., 2018). TOST is consistently lower than most satellite products and the model ensemble. Despite the spread in mean value, the models and observations largely agree in the magnitude of the increasing trend following 1997 ($0.82 \pm 0.13$ Tg yr$^{-1}$ in the CMIP6 ensemble vs. $0.70 \pm 0.15$ Tg yr$^{-1}$ in TOST vs. $0.83 \pm 0.85$ Tg yr$^{-1}$ in the satellite ensemble).

The right two panels of Fig. 7 demonstrate the sensitivity of the tropopause burden to the definition of the tropopause applied. Earlier model intercomparison studies generally utilized a chemical tropopause defined at the 150 ppbv ozone isopleth, since most models did not archive TCO calculated as an online diagnostic or tropopause pressure, and there is no clear tropopause

definition for tracers. However, there is a relatively large amount of ozone by mass in the upper troposphere, and the local column and global burden is sensitive to the exact definition applied. Model groups taking part in the CMIP6 experiments were asked to archive both monthly mean tropopause pressure as well as monthly mean TCO as calculated online with the dynamically varying tropopause and ozone concentrations. We calculate the tropospheric ozone burden using the monthly mean tropopause pressure in two different ways: first, excluding the mass of ozone in the layer containing the tropopause

(as commonly implemented; "exclusive"; yellow); and second, including the mass of ozone between the bottom of the layer containing the tropopause and the tropopause itself ("inclusive"; orange). The ozone mixing ratio in the layer containing the tropopause reflects a mixture of tropospheric and stratospheric air, and may be biased toward the higher stratospheric values. However, there is a potentially non-negligible amount of tropospheric ozone mass in this level, as reflected in the difference between the inclusive and exclusive calculations of the tropospheric burden in Fig. 7b-c. Either way, the inter-model spread in

tropospheric burdens is much higher when calculated with the pressure tropopause than the chemical tropopause (red). This is because there is large inter-model variability in the tropopause pressure (Fig. 2), and because the chemical tropopause by definition somewhat limits the amount of ozone mass in the troposphere. That being said, TCO calculated using the monthly mean chemical tropopause ends up being most similar in mean and variability to the online TCO diagnostic in the three models (GFDL-ESM4; MRI-EMS2-0; UKESM1-0-LL) that archived it using the dynamically-varying online pressure tropopause and

ozone (orange-red). In this study, we elect to use the exclusive pressure tropopause definition for defining the tropopause for purposes of the following budget calculations, but recommend future studies archive and explore the sensitivity of results to multiple definitions of the tropopause, especially with online TCO diagnostics.

## 4    Evolution of tropospheric ozone burden and budget over the period 1850-2100

### 4.1    Evolution of tropospheric ozone burden from 1850 to 2100

Figure 8 shows the evolution of the tropospheric ozone burden for the five models together with the multi-model mean. The burdens were calculated using the exclusive pressure tropopause definition, as discussed above, using the o3 variable defined in the AERmon CMIP6 table, on native model grids, and using the WMO tropopause pressures as archived in the ptp variable. All models show an increased burden over the period 1850-2100, with the largest rate of increase seen in the second half of the 20$^{\text{th}}$ century, and a decreased rate in the second half of the 21$^{\text{st}}$ century in response to declining emissions of ozone precursors.

The figure shows a large increase in tropospheric ozone burden, consistent with the increase in emissions of ozone precursors from the pre-industrial (PI) to the present day period (PD). The burden increases by 109 Tg from the PI (MMM 247 $\pm$ 36 Tg) to the PD (356 $\pm$ 31 Tg), with the most rapid change to burden occurring between 1950 and 1990. Figure 8 shows that the burdens calculated in CMIP6 models are consistent with those from ACCMIP time slice experiments for 1850, 1930, 1980 and 2000. There is good agreement between the two data sets, with a similar range in calculated model burden.

Good agreement is seen between the CMIP6 multi-model mean burden and separate estimates from TOAR (336 $\pm$ 8 Tg) derived from observational estimates of the whole-troposphere ozone burden using IASI and TOST data) for year 2000. The CMIP6 burden for the period 1990-2014 is however significantly higher than the TOST burden data presented above (section

3.4) for the same period. The origin of this discrepancy is not yet clear, and may emerge as more models with varying ozone distributions and tropospheric extent become available. Despite the high model bias with respect to these observational data, it is clear that a similar trend is observed for both model and observations, with both the TOST-derived burden and the CMIP6 historical mean burden increasing by around 15 Tg over the period 2000-2015. Further observational constraints are provided by the study of (Yeung et al., 2019) who used isotope data to estimate that the change in tropospheric ozone burden was no more than 40 % over the period 1850-2014, as well as the TOAR analysis that concluded a change in surface ozone concentrations of 32-71 % over this period (Tarasick et al., 2019). In CMIP6, the change in MMM, from 283 Tg to 356 Tg over this period, i.e. a change of 25 %, is consistent with this constraint.

The evolution in burden from 2014 to 2100 is shown for the ssp370 scenario. The burden increases by a further 60 Tg over the period 2015-2100. The major ozone precursors are projected to increase in the early part of ssp370 up to 2030 before beginning to level off after 2050 as in Figure 1. As anthropogenic $NO_x$ and CO emissions in ssp370 are projected to stabilise, the continued increase in ozone burden indicates an increasingly significant role for other ozone precursors, such as methane, which continues to increase until 2100 in this scenario, BVOCs, which increase due to changing climate and $CO_2$, and a likely increase in stratosphere-to-troposphere transport of ozone.

Figure 8 shows that the ozone burden in GISS-E2-1G shows the strongest response to increasing emissions, and consequently after approximately 1950, the largest tropospheric ozone burden. While the source of this strong increase in ozone burden with respect to other models is difficult to attribute, Kelley et al. (2020) note that there is a significant bias in the stratospheric ozone column, which is also connected to a positive tropospheric ozone column bias, and a too-cold tropopause in this model configuration.

The response of UKESM1 is more muted, with UKESM1 showing the largest ozone burden in 1850 of all models and into the early stages of the simulations, which is consistent with the largest LNOx and BVOC emissions. The present day ozone burden is affected by a strong decrease in downward transport of ozone from the stratosphere (Skeie et al., 2020), as discussed below. Together, these two factors contribute to the increase in ozone burden from 1850 to 2014 being the smallest in this model.

The range in simulated burden varies little across the historical period in the five simulations, being 36 Tg in PI conditions, with UKESM1 showing the highest burden, and 31 Tg for 2005-2015, with GISS-E2-1-G the highest.

## 4.2 Regional changes

Figures 9-10 show the historical changes in tropospheric ozone distribution in the CMIP6 ensemble since the preindustrial. Over the historic period, ozone increases throughout the troposphere, with greatest increases occurring in the NH. The largest relative changes occur near the surface in the NH, especially downwind of eastern North America and East Asia, where the rise in ozone precursor emissions (Fig. 1) were predominantly located. Of the three periods explored, the bulk of the increase in ozone occurred between the 1930s and 1980s. Since the 1980s, most of the increases were located in South and East Asia, and the southern tropics and subtropics, reflecting the implementation of aggressive precursor emission controls in North America

and Europe. The recent increase of ozone in South and East Asia has led to an increase in the inter-model spread of ozone relative to earlier periods and other regions.

Figures 11-12 show the future changes in tropospheric ozone distribution in the CMIP6 ensemble relative to the present day. Future changes are expected to be less dramatic than the 1850 to 2014 increase, reflecting the reduction in $NO_x$ emissions and relative stabilization of CO emissions in the ssp370 scenario (Fig. 1). Despite the global non-methane precursor emission reductions, tropospheric ozone still increases across the 21$^{st}$ century possibly driven by a combination of enhanced stratospheric downwelling associated with a GHG-driven acceleration to the BDC, increasing methane in the ssp370 scenario (cf. lower right panel of Fig. 1) and increasing tropopause height. In particular, the increase in the subtropical upper troposphere likely reflects the influence of increased stratospheric downwelling coupled with stratospheric ozone super-recovery (c.f. Fig. 10, Keeble et al., 2020). The models predict that TCO decreases over the remote Pacific, likely reflecting precursor emission reductions coupled with a temperature-driven increase in ozone-destroying tropospheric water vapour.

### 4.3 Global ozone budget

We report here data for models that diagnosed the required chemical ozone production (o3prod), chemical ozone loss (o3loss) and ozone dry deposition (dryo3) outputs for both the Historical and ssp370 experiments. Figure 13 shows the evolution of globally integrated annual mean ozone dry deposition (DD), net chemical ozone production (NCP $=P_{O3}$ - $L_{O3}$), and the inferred net stratospheric to tropospheric transport (STE: derived as the "residual" in the ozone budget, i.e., "Residual = o3loss - o3prod + dryo3"). For this analysis, we used the CMIP6 data request for ozone production and loss: $P_{O3}$ is defined as the sum of reaction tendencies through $HO_2$/$CH_3O_2$/$RO_2$ + NO reactions, and $L_{O3}$ as the sum of O($^1$D)+$H_2O$, $O_3$+$HO_2$ and OH, and $O_3$+alkenes. The tropospheric ozone budget terms, burden and lifetime for the historical and future ssp370 simulations are reported in Tables 1 and 2, respectively. The results are averaged over 10 years for each period. As with the tropospheric burden calculation, we used monthly mean output for each variable and the WMO tropopause definition to define the limit of the tropopause using monthly mean output and mask the reaction tendency data accordingly.

For the Historical and ssp370 coupled experiments, the GISS-E2-1-G model did not provide the chemical loss term (L), and so we only include its production (P) and dry deposition (DD) in the tables (1 and 2). A notable feature is that P and DD from the GISS-E2-1-G model are significantly (at least 50 %) higher than similar data for the other models reported here. In light of the good agreement between models in terms of ozone abundance, this is somewhat surprising given that the higher production is offset by a similarly fast ozone deposition at the surface, giving ozone burden and abundance that agree reasonably with other models.

Figure 13 shows the global total dry deposition tendency for ozone in Tg yr$^{-1}$. Global total deposition increases over the period 1850 to 2100 for all models, increasing gradually until the 1950s before increasing more steeply until the late 1990s. The variation in dry deposition largely reflects the evolving ozone burden which increases over the PI to PD period and stabilises from PD into the later 21$^{st}$ century. Excluding GISS-E2-1-G, there still remain significant differences in ozone dry deposition among the models before the 1950s (e.g., from 460 Tg yr$^{-1}$ in GFDL-ESM4 to 633 Tg yr$^{-1}$ in UKEMS1 for 1850s), but the differences are smaller after the year 2000 (815$-$907 Tg yr$^{-1}$ over 2005-2014).

Figure 13 shows a more complex behaviour in NCP. There is a small increase in ozone production over the period 1850−1950, at which point there is a more rapid rise in the emission of tropospheric ozone precursors and hence burden, see Figure 1. This rapid increase continues until around 1980 at which the growth in emissions slows. The projected emissions, and NCP, reach a maximum between 2030 and 2050, and subsequently stabilise.

       GISS-E2-1-G is erroneously missing the loss of ozone with isoprene and terpenes in its reported o3loss variable, making the
net chemical production term erroneously high. When the online calculation of the stratosphere-to-troposphere flux of ozone is used instead to calculate the net chemical production term in that model (not reported to ESGF, but obtained from the original simulations), the temporal evolution of the net production term is qualitatively consistent with the other four models. While each term in the GISS-E2-1-G ozone budget has a larger magnitude than the other models, these components nevertheless sum to create ozone mixing ratios and burdens comparable to, albeit still larger than, the other models and the observations.

The other four models show similar behaviour across time with NCP peaking around 2030 but different absolute responses to the increase in emissions, with the PI to PD change in NCP being 585 Tg yr$^{-1}$ for UKESM1, compared to 460 Tg yr$^{-1}$ for CESM2-WACCM, 400 Tg yr$^{-1}$ for GFDL-ESM4 and 353 Tg yr$^{-1}$ for MRI-ESM2-0. In 1850-1859, after the GISS-E2-1-G NCP of 1500 Tg yr$^{-1}$, UKESM1 shows the highest NCP, of around 250 Tg yr$^{-1}$, while the other three models show much smaller NCP around 60 Tg yr$^{-1}$, which is similar to values reported for 1900 in (Wild and Palmer, 2008). The higher NCP in
UKESM1 is consistent with higher LNOx and BVOC emissions in the early part of the historical period, compared to other three models.

       Figure 14 shows the variation in vertically integrated zonal mean net chemical tropospheric ozone production over the period 1850 to 2100. GISS-E2-1-G is excluded from this plot for reasons discussed above. In the 1850s, the main region of ozone production is located in the tropics and arises from emissions of $NO_x$ due to biomass burning at the surface and $NO_x$
production in the UT from lightning. Over the period 1850-2100, an increase in net ozone production in the mid-latitudes of NH is observed for all models. In the 20[th] century, ozone production can be seen to commence in NH mid-latitudes in response to the increase in anthropogenic emissions in these regions. There is a substantial increase in the extent of regions of strong, positive NCP in the NH extratropics from the mid 20[th] century onwards, and some expansion of the region of positive NCP into the southern subtropics can be seen beginning around 1980. There is good agreement between the models on these points,
but there are some interesting regional differences in this period that merit further study: UKESM1 shows net positive ozone tendency throughout the NH across the whole historical period, in contrast to the other models, and CESM2-WACCM and GFDL-ESM4 show net ozone destruction in high latitude regions in contrast to UKESM1 and MRI-ESM2-0. The figure shows that, as in Figure 13, around the year 2010, NCP reaches a maximum and then begins to decline, presumably in response to the projected decrease in emissions of tropospheric ozone precursors in the later part of the 21[st] Century (Revell et al., 2015).

The models also agree in simulating net ozone destruction across the mid-latitudes of the SH, due to a combination of low emissions and chemical ozone destruction via ozone photolysis and reaction with $HO_x$ radicals in the free troposphere and over the oceans (Cooper et al., 2014). Ozone destruction in this region reaches a minimum around 2000, presumably due to a shift in emissions southward during the later 20[th] century (Zhang et al., 2016). In the 21[st] century, there is a pronounced increase in ozone destruction in the SH tropics, reflecting a warmer and wetter future climate that promotes ozone chemical destruction

through the reaction of O($^1$D) and H$_2$O following ozone photolysis (Stevenson et al., 2006) and higher concentrations of HO$_x$ radicals (Doherty et al., 2013; Johnson et al., 1999). In the tropics, there is a strong net ozone destruction in CESM2-WACCM over the whole period, with an increase towards the end of 21$^{st}$ century; this tropical feature is much weaker in the other three models and there is even slightly net positive ozone production in UKESM1 before around 2020.

Figure 15 shows that both chemical production and loss terms, $P_{O3}$ and $L_{O3}$, increase over the 20$^{th}$ century, albeit with terms that increase at different rates over the period. The chemical production increases rapidly over the 20$^{th}$ century, particularly in GISS-E2-1-G, CESM2-WACCM and UKESM1, and the rate of increase slows in the 21$^{st}$ century as projected emissions reductions begin to have an impact. Chemical destruction also increases over the entire period, largely following ozone burden increases, but also reflecting increases in HO$_x$ radicals, as discussed above. After 2030, the destruction rate increases faster than production, and NCP begins to decrease. The steadily increasing ozone burden in all models, despite the declining NCP in four of the five, demonstrates the increasingly large role of downward transport of ozone from the stratosphere to the ozone burden in the later part of this century.

Ozone production efficiency (OPE) (Liu et al., 1987), defined as moles of ozone produced per mole of NO$_x$ emitted, is included here as a way to compare the different model ozone responses to changes in NO$_x$ emissions. It can also be compared to OPE derived from in-situ measurements of O$_3$ and NO$_x$, e.g. Travis et al. (2016). For the experiments presented here, OPE was calculated from o3prod and eminox/emilnox variables, so as to include NO$_x$ from anthropogenic, biological and lightning sources. We present the OPE as a way of comparing model responses to a given change in NO$_x$ emissions across the period 1850-2100. By normalising for the important driver of NO$_x$ emissions, the OPE illustrates the variation between models of the chemical response to emissions changes, which can arise from the differing treatment of processes such as photolysis, deposition, transport and mixing. Figure 16 shows the ozone production efficiency for the five models. The five models show similar behaviour, with OPE declining from large initial values to a minimum over the period 1980-2050 before recovering in the late 21st century. The trend suggests that models respond less sensitively to NO$_x$ emissions as the tropospheric NO$_x$ burden increases, with the OPE mirroring somewhat the NO$_x$ burden plots of Figure 1. Throughout the period 1850-2100, the OPE of the GISS-E2-1-G model is significantly higher than that of the other models, consistent with the higher ozone production and with the stronger response of ozone to surface NO$_x$ emissions noted in (Wild et al., 2020). The OPE for the other models is lower, and more similar, indicating that the models' chemistry have similar ozone responses to increases in NO$_x$ levels. The OPE recovers somewhat in the 21$^{st}$ century, during which time ozone production responds more sensitively to increasing NO$_x$, with implications for air quality control measures. As the OPE is a function of the background NMVOC mixing ratio, the higher VOC emissions in the period 1850-1900 in UKESM1 appear to account for the higher OPE. Similarly, the higher OPE of CESM2-WACCM at the end of the 21$^{st}$ century is likely to be the result of the higher biogenic VOC emissions in this model.

Based on calculated ozone burden and rates of ozone removal, the ozone lifetime defined as B/(L+DD) decreases across the historical period and into the 21st century. In 1850, the multi-model mean ozone lifetime is $29.5 \pm 2.1$ days, decreasing by 4 days to $25.5 \pm 2.2$ days in the present day. This decrease continues in the 21$^{st}$ century to $23.2 \pm 2.7$ days in 2100. The decrease in lifetime is driven partly by an increase in L, responding to the higher temperatures and humidity impacting the rates of ozone

destruction reactions (Young et al., 2013), and partly by increasing DD, which is a response to increasing ozone concentration at the surface. Together these offset the increase in lifetime which would be calculated from an increase in ozone burden.

Tables 1 and 2 show the residual term in the ozone budget. Again, it should be noted that the data for GISS-E2-1-G are not directly comparable with the data from the other models, being instead an integrated dynamical ozone flux across the tropopause. While there is a large inter-model spread in NCP and dry deposition terms (i.e., substantially higher values in
UKESM1), there are similar residual terms in the ozone budget (i.e., the inferred net stratospheric influx) before the 1950s of between 400 and 500 Tg per year. These values decrease sharply after 1970 partly due to the effect of stratospheric ozone depletion by amounts ranging from 300 Tg (UKESM1) to 60 Tg (GFDL-ESM4). This decline in residual is a robust feature across models and is consistent with reduced ozone STE in the present-day compared to pre-industrial times as a result of stratospheric ozone depletion despite an acceleration of the stratospheric residual circulation, and a potential increase in the
troposphere-to-stratospheric flux of ozone in the northern latitudes (c.f. Fig. 9i-j). After the year 2000, the residual terms starts to increase in all models coinciding with the expected ozone recovery, as ozone depleting substances decrease, and the BDC increase, resulting from increasing GHGs. This is in line with recent studies using CCMs including a stratospheric ozone tracer which provide evidence that both the acceleration of the BDC and stratospheric ozone recovery will tend to increase the future global tropospheric ozone burden through enhanced STE, with the magnitude of the change depending on the RCP scenario
(Banerjee et al., 2016; Meul et al., 2018; Akritidis et al., 2019). This projected increase in STE associated with climate change and ozone recovery offsets decreases in net chemical production associated with reductions in ozone precursor emissions, in agreement with e.g. Sekiya and Sudo (2014).

Models differ in their simulations of stratospheric ozone, which inevitably affects tropospheric ozone through stratosphere-troposphere coupling. Figure 17 shows preindustrial zonal mean ozone (PI: averaged over 1850-1859), changes in ozone between the PI and the present-day periods (PD; averaged over 1995-2004), and the change between PD and the end of the 21$^{st}$
century (2090-2100) in all five models. In the PI case, UKESM1 has the largest ozone mixing ratios throughout the troposphere among the five models, which is associated with its large ozone production (Figure 14) and net ozone production (Figure 15). Figure 17 shows that the GISS model shows a higher tropopause than other models, which gives a higher tropospheric ozone burden for GISS-E2-1-G (Tables 1 and 2).

The propagation of ozone from the stratosphere to the troposphere is evident in all five models and this influx of ozone, to a
610 varying extent, contributes to the tropospheric ozone burden. Figure 17 shows strong stratospheric ozone depletion in UKESM1 between 1850 and 2014 which results in a reduced net input of stratospheric ozone into the troposphere. This strong ozone depletion is consistent with the very low residual budget term for this model in the present day. In Figure 17 the tropopause height is shown to increase across all models over the historical period at southern high latitudes due to the circulation changes
associated with increasing GHGs and ozone depletion, but with a smaller but still visible increase in the NH mid-latitudes in UKESM1. Over this period there are substantial ozone increases in the high-latitude NH lower stratosphere, which would also enhance stratosphere-to-troposphere transport of ozone, in all models except UKESM1. Despite the larger increase of NCP in UKESM1 (from 279 to 830 Tg yr$^{-1}$ compared to an increase from 78 to 530 in CESM2-WACCM, from 86 to 466 in GFDL-ESM4 and 58 to 411 Tg yr$^{-1}$ in MRI-ESM2-0; Table 1), the decrease in transport of ozone from the stratosphere results in

UKESM1 showing a smaller increase in ozone burden increase from 1850 to 2014, as noted elsewhere (Keeble et al., 2020; Skeie et al., 2020). From the PD into the future, all models show pronounced stratospheric ozone increases, which visibly impact the tropospheric ozone abundance. Again, UKESM1 shows the smallest increase in tropospheric ozone among the five models, which may be linked to the calculated decrease of ozone near the tropopause that could be linked to the increase in the tropopause height in future climate. However, such behaviour is not obvious in the other models which also show a slight increase of the tropopause height.

More detailed study of the influence of the stratosphere on the troposphere is difficult in the context of CMIP6 and its data request. While well-described in the literature (Holton et al., 1995; Appenzeller et al., 1996; Jaeglé et al., 2017), in the CMIP6 data request there is no diagnostic output for dynamical transport of ozone across the tropopause, and so the residual method has to be employed as in previous Assessments (Stevenson et al., 2006; Young et al., 2018). This is an acceptable method provided the overall ozone tendency is small (Hu et al., 2017), and has been shown to give good agreement with the dynamical STE for models such as UKESM1 (Griffiths et al., 2020). The values determined here using the residual method agree reasonably with direct calculation of STT using various tropopause definitions of 410–450 Tg/yr (Yang et al., 2016) and the range of 400–500 Tg given in (Olsen et al., 2013). The supplementary section shows comparisons between dynamical STT calculations using a subset of the models described here using an online tropopause, and shows that there is good agreement between the dynamical calculations and the residual method.

## 5  Summary and conclusions

We have analysed the evolution of tropospheric ozone in CMIP6 CMIP Historical and ScenarioMIP ssp370 experiments, a "regional rivalry" pathway. Ozone has been evaluated against a broad range of observations spanning several decades, and we have determined the evolution of the tropospheric ozone burden over the period 1850-2100. For this analysis, we have concentrated on coupled atmosphere-ocean experiments using whole atmosphere chemistry and interactive ozone. We excluded those models that use simplified chemistry which have been shown to yield low ozone burdens, with the availability of data limiting us to an analysis of ozone burden in five models and the ozone budget for four models.

We evaluated these CMIP6 models against a suite of surface, sonde and satellite products for the recent past. The models tend to overestimate ozone in the northern hemisphere and understimate ozone in the southern hemisphere. Nevertheless, the models well-reproduce the spatial and seasonal variability in the tropospheric ozone distribution, and capture the observed increasing trends in tropospheric ozone since at least 1998.

However, a key uncertainty identified by this analysis regards the definition of the troposphere. We compared definitions based on the chemical tropopause (as traditionally applied) versus the pressure tropopause and online tropospheric ozone diagnostics. All three varied significantly from one another, and we recommend future model inter-comparison studies explicitly examine the sensitivity of results to tropopause definition applied, including an emphasis toward online tropospheric ozone column calculations.

The ozone burden grows by 44 % from PI (247 ± 36 Tg) to the PD (356 ± 31 Tg), and reaches a maximum of 416 ± 36 Tg in 2100. The inter-model range is roughly constant across the integration, being around 8 %.

The ozone budget has been analysed in terms of ozone chemical production, loss, deposition, and the STE. Deposition, chemical ozone production and loss have been shown to increase steadily from the PI into the future, with the evolution of the ozone burden likely moderated by the behaviour of the stratospheric ozone burden (e.g., Morgenstern et al., 2018), lightning $NO_x$ and global methane abundances, despite any reductions in non-methane precursor emissions. The variation in the growth rate of the ozone burden is shown to depend sensitively on the growth rate of emissions and the STE. There remains wider diversity between modelled ozone budget terms, with UKESM1 showing the largest tendencies, particularly in net chemical production, and the smallest STE.

At the start and end of the model period, inter-model diversity appears to be affected by differences in emissions of biogenic VOCs and LNOx. In contrast to the prescribed anthropogenic $NO_x$ and CO emissions, emission fluxes of BVOCs are calculated online, as a function of environmental parameters. There is considerable variation in BVOC emissions across the models, and in the PI, UKESM1, the model with the highest ozone burden, has the largest emissions of BVOCs. The sensitivity of ozone production to $NO_x$ emissions has been calculated in the form of ozone production efficiency. There is much greater similarity between models in this case, reflecting similar sensitivities in the underpinning chemical mechanisms, although only four of the models show very similar OPE. The higher OPE in GISS-E2-1-G reflects its greater sensitivity to emissions, resulting in a larger production term, a larger source of lightning $NO_x$ than the other models, and a shorter $NO_x$ lifetime. The OPE, which is large in the PI, reaches a minimum around the PD, before recovering again into the later part of the 21$^{st}$ century.

The impact of the stratosphere on tropospheric ozone burden has been demonstrated. We find that the residual STE tendencies are similar among the models in the PI, but that the STE evolves differently in the five models: UKESM1 has the largest ozone depletion in both hemispheres, whereas in CESM2-WACCM, MRI-ESM2-0 and GFDL-ESM4 there are ozone increases in the lower stratosphere northern high latitudes; this goes along with the inferred STE being very low in UKESM1 which may contribute to the smallest ozone burden trend in this model. Differences in stratospheric ozone in the models contribute significantly to the model spread in diagnosing ozone budget. GISS-E2-1-G is again an outlier in terms of its behaviour, with STE increasing across the period 1850-2100, presumably the result of tropospheric expansion.

Stratospheric ozone depletion and recovery to tropospheric ozone has the biggest effect on the budget calculations around the year 2000. In this period, the decline in stratospheric ozone, and presumably STE, offsets a significant increase in net chemical ozone production over the period 1980-2000, which partially mitigates the response of tropospheric ozone to rapidly increasing emissions. The tropospheric burden over this period is therefore lower than it might otherwise have been, although the precise level of offset requires further clarification.

There remains a need to assess these future changes at the regional scale, and to understand which regions of the troposphere are most affected by future stratospheric ozone changes.

Looking forward, there is a clear need to improve the diagnostic data request for the evaluation of tropospheric ozone budgets, especially for multi-model intercomparison studies. The closure of the ozone budget remains problematic due to missing terms in the o3prod, o3loss and dryo3 diagnostics (e.g., photolysis of nitrates, deposition of $NO_y$ species) being of

similar magnitude to the residual terms, and their absences introduce large uncertainties to the budget calculations. We would propose that a consistent odd-oxygen family first be defined that accounts for ozone and its fast cycling with $NO_x$ and its reservoirs (e.g., Wang et al., 1998; Bates and Jacob, 2019), for which the net chemical tendency (i.e. $P-L$) may then be easily calculated by comparing total family member mass before and after each call to the chemical operator. This will guarantee that all relevant chemical reactions are included regardless of each model's different chemical mechanism and minimize the chance of coding errors. The net odd-oxygen deposition tendency may then be similarly determined across the dry and wet deposition operators. We also recommend that online ozone and odd-oxygen mass fluxes across the tropopause be diagnosed and archived to compare to the residual method from which one may evaluate budget closure. Lastly, a consistent definition of what mass is considered tropospheric should be defined; of the possibilities, we recommend the "inclusive" definition as the most physical and appropriate.

*Data availability.* Data used for this study were obtained from the Earth System Grid Federation (http://esgf-node.llnl.gov) on October 23, 2020. Data for the surface ozone data are available at ftp://aftp.cmdl.noaa.gov/data/ozwv/SurfaceOzone/ and https://www.gaw-wdcrg.org.

*Author contributions.* PTG, LTM, GZ, PJY, YMS, IEG, DT, JL, OM, ST, MD and NO provided data analysis and contributed to the writing and discussion of this manuscript. VN, PZ, ATA, LWH, OW, JMK, FOC, BH, STT, contributed to the writing and discussion. STT compiled the data in the Supplementary Table. PTG, FOC, LTM, VN provided data and data analysis for Figure 1 of the Supplementary, for which NLA contributed to the preparation of UKESM1 experiments. GZ prepared the data for Figure 2 of the Supplementary.

*Competing interests.* The authors declare no competing interests.

*Acknowledgements.* This work used JASMIN, the UK collaborative data analysis facility. PTG, ATA, JMK and NLA thank NCAS and the Met Office for funding and support of the UKCA project. FMOC and BH were supported by the European Union's Horizon 2020 Framework Programme for Research and Innovation "Coordinated Research in Earth Systems and Climate: Experiments, kNowledge, Dissemination and Outreach (CRESCENDO)" project under Grant Agreement No. 641816. GZ was supported by the NZ Government's Strategic Science Investment Fund (SSIF) through the NIWA programme CACV. MD and NO were supported by the Japan Society for the Promotion of Science (grant numbers: JP18H03363, JP18H05292, and JP20K04070), the Environment Research and Technology Development Fund (JPMEERF20172003, JPMEERF20202003, and JPMEERF20205001) of the Environmental Restoration and Conservation Agency of Japan, and the Arctic Challenge for Sustainability II (ArCS II), Program Grant Number JPMXD1420318865. We acknowledge the World Climate Research Programme, which, through its Working Group on Coupled Modelling, coordinated and promoted CMIP6. We thank the climate modeling groups for producing and making available their model output, the Earth System Grid Federation (ESGF) for archiving the data and providing access, and the multiple funding agencies who support CMIP6 and ESGF.

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

| Historical | | UKESM1 | CESM2-WACCM | GFDL-ESM4 | MRI-ESM2-0 | GISS-E2-1-G |
|---|---|---|---|---|---|---|
| 1850-1859 | P | 3409 | 2225 | 2291 | 2271 | 4311 |
| | L | 3155 | 2155 | 2225 | 2212 | – |
| | P-L | 254 | 70 | 66 | 58 | – |
| | DD | 633 | 459 | 471 | 549 | 1000 |
| | Residual | 379 | 387 | 404 | 491 | **1878**[*] |
| | Burden | 291 | 204 | 221 | 248 | 272 |
| | Lifetime | 27.7 | 28.2 | 29.5 | 32.4 | – |
| 1895-1904 | P | 3492 | 2331 | 2418 | 2367 | 4464 |
| | L | 3212 | 2253 | 2332 | 2297 | – |
| | P-L | 279 | 78 | 86 | 70 | – |
| | DD | 654 | 481 | 497 | 574 | 1051 |
| | Residual | 374 | 403 | 410 | 504 | **1872**[*] |
| | Burden | 298 | 211 | 229 | 256 | 279 |
| | Lifetime | 27.8 | 27.9 | 29.1 | 32.1 | – |
| 1945-1954 | P | 3922 | 2807 | 2921 | 2798 | 5457 |
| | L | 3522 | 2628 | 2734 | 2631 | – |
| | P-L | 400 | 179 | 187 | 167 | – |
| | DD | 730 | 579 | 611 | 675 | 1336 |
| | Residual | 329 | 400 | 424 | 508 | **1962**[*] |
| | Burden | 318 | 239 | 260 | 285 | 315 |
| | Lifetime | 26.9 | 26.8 | 28.0 | 31.0 | – |
| 1975-1984 | P | 4677 | 3699 | 3822 | 3560 | 6691 |
| | L | 4004 | 3277 | 3440 | 3201 | – |
| | P-L | 673 | 422 | 382 | 359 | – |
| | DD | 837 | 725 | 774 | 844 | 1759 |
| | Residual | 164 | 303 | 392 | 485 | **2005**[*] |
| | Burden | 345 | 282 | 307 | 334 | 355 |
| | Lifetime | 26.6 | 26.8 | 24.6 | 27.4 | – |
| 1995-2004 | P | 5315 | 4366 | 4371 | 3987 | 8377 |
| | L | 4476 | 3835 | 3905 | 3576 | – |
| | P-L | 839 | 530 | 466 | 411 | – |
| | DD | 867 | 791 | 833 | 892 | 1992 |
| | Residual | 28 | 261 | 367 | 481 | **1991**[*] |
| | Burden | 354 | 310 | 327 | 357 | 387 |
| | Lifetime | 23.8 | 24.1 | 25.4 | . 28.7 | – |

**Table 1.** Tropospheric ozone budget terms for the three models averaged over each 10-year historical period. P for chemical production, L for chemical loss, P−L for net chemical production, DD for dry deposition, and Residual is the term balance by Residual=L-P+DD. Units of "P", "L", "DD", and "Residual" are in $Tg(O_3)yr^{-1}$, "Burden" in $Tg(O_3)$, and "Lifetime" in days. The Residual quantities for GISS-E2-1-G was calculated differently from the others, being based on dynamical transport rather than budget closure, and so is indicated in bold with an asterisk.

| SSP370 | | UKESM1 | CESM2-WACCM | GFDL-ESM4 | MRI-ESM2-0 | GISS-E2-1-G |
|---|---|---|---|---|---|---|
| 2025-2034 | P | 5867 | 4996 | 4805 | 4327 | 9106 |
| | L | 4977 | 4399 | 4330 | 3905 | – |
| | P-L | 890 | 597 | 475 | 422 | – |
| | DD | 894 | 863 | 879 | 937 | 2150 |
| | Residual | 4 | 266 | 404 | 515 | **2318**$^*$ |
| | Burden | 373 | 346 | 355 | 381 | 439 |
| | Lifetime | 22.9 | 23.6 | 24.6 | 28.3 | – |
| 2045-2054 | P | 6114 | 5311 | 4974 | 4498 | 9434 |
| | L | 5273 | 4756 | 4535 | 4112 | – |
| | P-L | 841 | 555 | 439 | 386 | – |
| | DD | 899 | 895 | 898 | 952 | 2178 |
| | Residual | 58 | 340 | 459 | 566 | **2546**$^*$ |
| | Burden | 386 | 364 | 371 | 393 | 468 |
| | Lifetime | 22.5 | 23.2 | 24.6 | 27.9 | – |
| 2090-2099 | P | 6763 | 5909 | 5324 | 4828 | 10350 |
| | L | 6089 | 5527 | 4981 | 4563 | – |
| | P-L | 675 | 382 | 343 | 266 | – |
| | DD | 887 | 904 | 898 | 957 | 2141 |
| | Residual | 212 | 522 | 555 | 692 | **2868**$^*$ |
| | Burden | 406 | 378 | 389 | 411 | 499 |
| | Lifetime | 20.9 | 21.2 | 23.8 | 26.8 | – |

**Table 2.** Same as Table 1 but for ssp370. As before, the Residual quantities for GISS-E2-1-G were calculated differently from the others, being based on dynamical transport rather than budget closure, and so is indicated in bold with an asterisk.

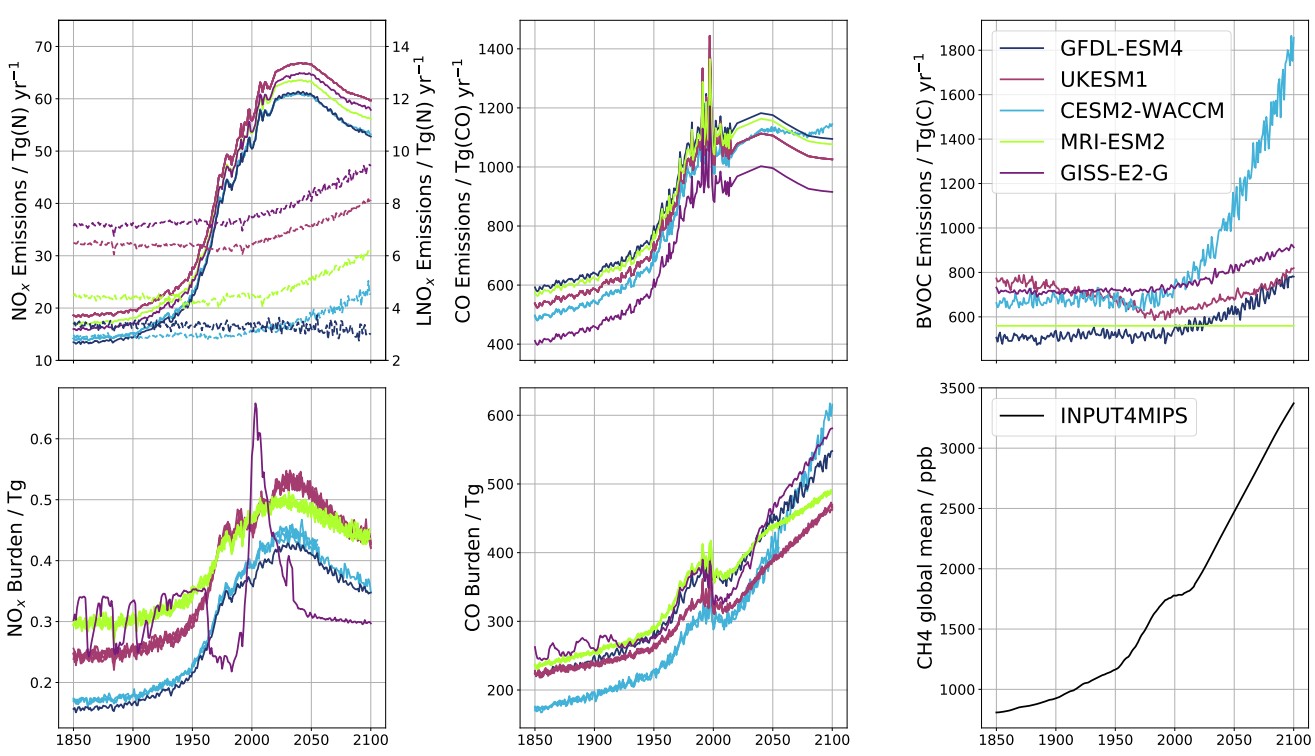

**Figure 1.** Diagnosed emissions and burden of tropospheric ozone precursors. Maroon line: UKESM1; Light blue line: CESM2-WACCM; Dark blue line: GFDL-ESM4; Dark red line: GISS-E2-1-G; Green line: MRI-ESM2-0.

.

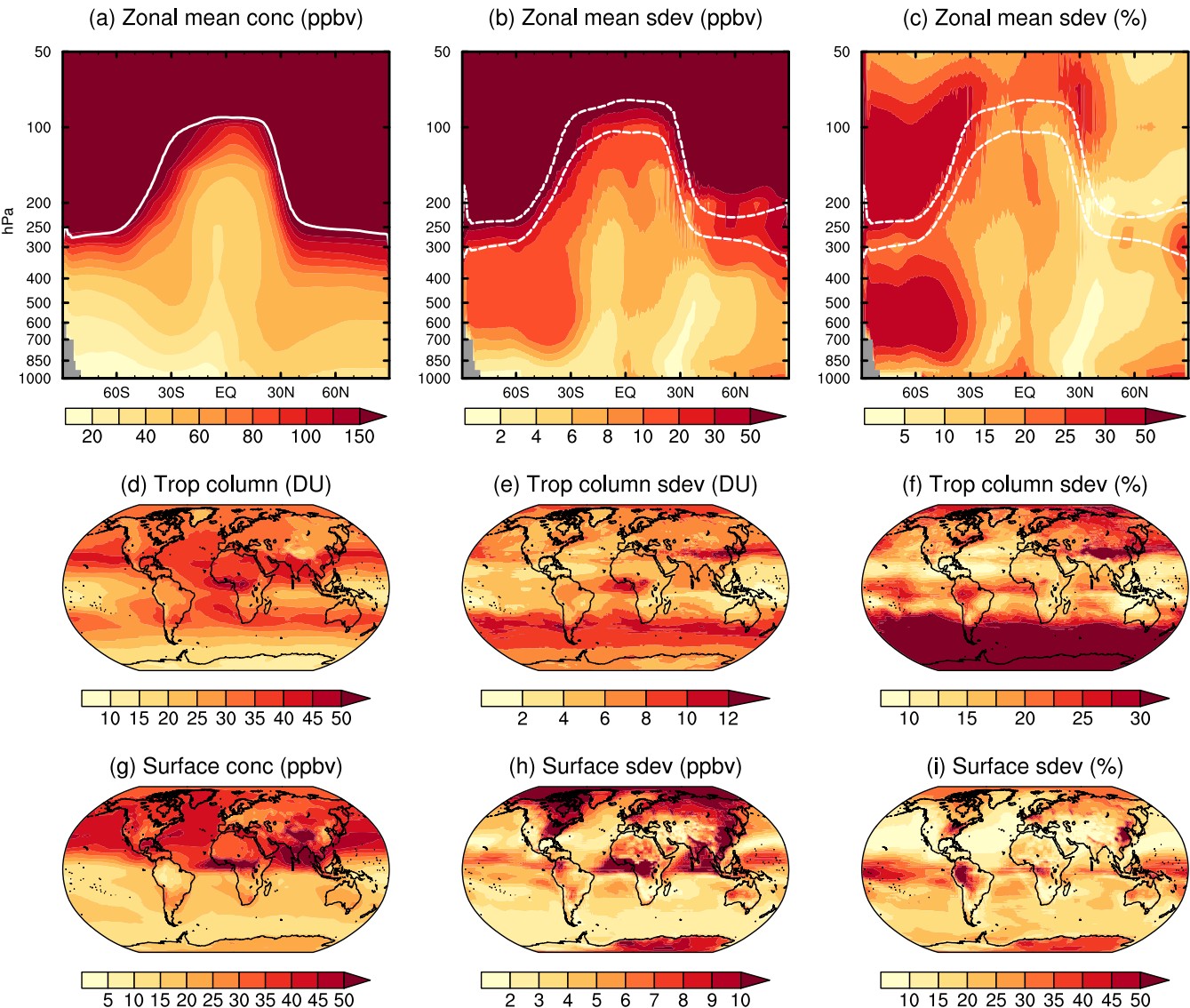

**Figure 2.** CMIP6 ensemble mean, annual mean ozone climatologies, and their inter-model variability in the present day (2005-2014 C.E.) of the historical simulation. The top row shows zonal mean ozone, the middle row shows the tropospheric ozone column, and the bottom row shows surface ozone. For each row, the left hand panel shows the absolute values of the ozone variable: ppbv for the zonal mean and surface concentrations, and Dobson units (DU) for the tropospheric column. The middle column shows the absolute inter-model standard deviations in the same units. The right column shows the standard deviation as a percentage of the ensemble mean value. The top row also shows the multi-model zonal mean tropopause pressure (left panel), and the mean ± one standard deviation of the multi-model variability (middle and right panels). Note that each panel has a different scale. This is an updated version of Fig. 3 of Young et al. (2013).

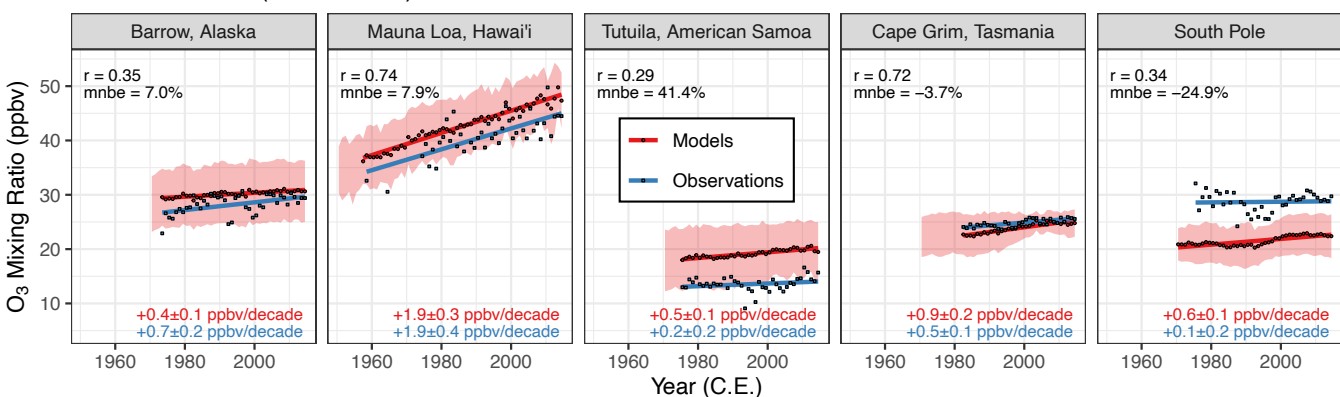

**Figure 3.** Comparison of annual mean surface observations with the multi-model mean at 5 stations: Barrow, Alaska, USA (71.3°N, 156.6°W, 11 m.a.s.l.), Mauna Loa, Hawai'i, USA (19.5°N, 155.6°W, 3397 m.a.s.l.), Cape Matatula, Tutuila, American Samoa (14.2°S, 170.6°E, 42 m.a.s.l.), Cape Grim, Tasmania, Australia (40.7°S, 144.7°E, 94 m.a.s.l.), and the South Pole (90.0°S, 59.0°E, 2840 m.a.s.l.). The models are sampled from the surface level, except for Mauna Loa, which is sampled at 680 hPa. The pink shading represents the multi-model mean and ± one standard deviation at each location. The red circles indicate the multi-model mean sampled at the month of the observations. The blue squares represent the observations. The solid lines show an ordinary least-squares regression for the multi-model mean and the observations, with the respective slope printed in the lower right of the panel. The temporal correlation (r) and mean normalized bias error (mnbe) are shown in black for each panel.

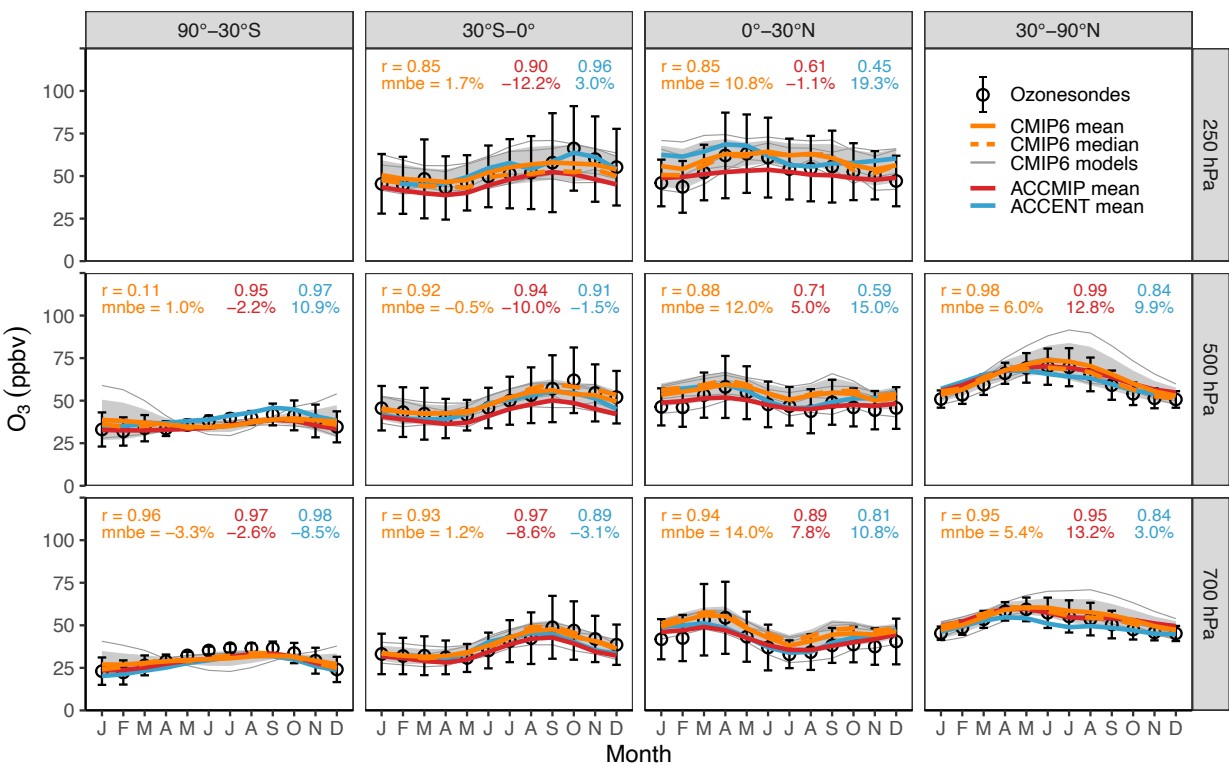

**Figure 4.** Comparison of the annual cycle of ozone, between ozonesonde observations (black circles) and the CMIP6 ensemble mean (solid orange line), CMIP6 ensemble median (dashed orange line), the ACCMIP ensemble mean (red line; Young et al., 2013) and the ACCENT ensemble mean (blue line; Stevenson et al., 2006). CMIP6 model data is from years 2005 to 2014 of the historical experiment. Model and observational data were grouped into four latitude bands (90°S to 30°S, 30°S to 0°, 0° to 30°N and 30°N to 90°N) and sampled at three altitudes (700 hPa, 500 hPa and 250 hPa), with the models sampled at locations and months of the ozonesonde measurements before averaging together. The individual CMIP6 models and ensemble members are represented by the thin grey lines, with the grey shaded area indicating ± 1 standard deviation about the CMIP6 ensemble mean. Error bars on the observations indicate the average interannual standard deviation for each group of stations. The correlation (r) and mean normalised bias error (mnbe) for the CMIP6 (orange), ACCMIP (red) and ACCENT (blue) ensemble means versus the observations are also indicated in each panel. This figure is an update of Fig. 4 of Young et al. (2013).

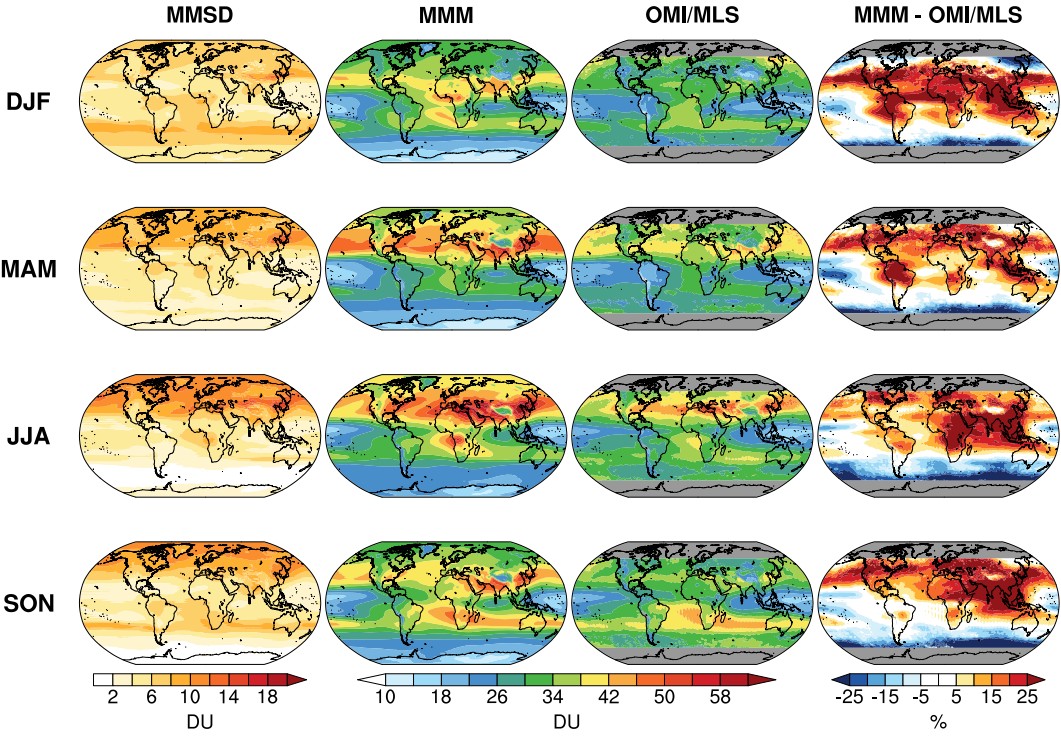

**Figure 5.** Comparison of the seasonal cycle of tropospheric column of ozone (TCO) abundances with satellite climatology for the period 2005 to 2014. Each row shows a separate meteorological season, from top to bottom: December to February (DJF), March to May (MAM), June to August (JJA), and September to November (SON). The left column shows the inter-model standard deviation of seasonal mean TCO in the CMIP6 ensemble in Dobson Units (DU). The second from the left column shows the multi-model seasonal mean TCO in DU. The second from the right column shows the seasonal mean TCO in the OMI/MLS product (Ziemke et al., 2006). The right column shows the relative bias in the multi-model seasonal mean relative to the OMI/MLS product in percent (%).

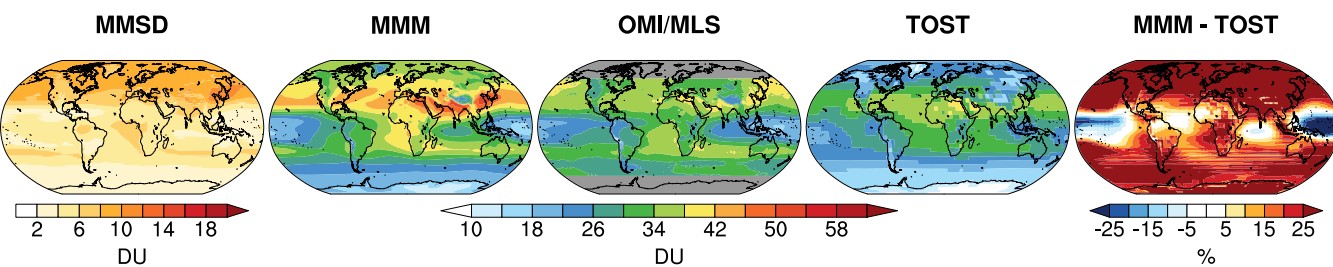

**Figure 6.** Comparison of the annual tropospheric column of ozone (TCO) abundance with satellite (OMI/MLS) and ozonesonde-derived (TOST) climatologies for the period 2005 to 2014. The left column shows the inter-model standard deviation of annual mean TCO in the CMIP6 ensemble in Dobson Units (DU). The second from the left column shows the multi-model annual mean TCO in DU. The middle column shows the annual mean TCO in the OMI/MLS product (Ziemke et al., 2006). The second from the right column shows the annual mean TCO in the TOST product (Liu et al., 2013b, a). The right column shows the relative bias in the multi-model mean relative to the TOST product in percent (%).

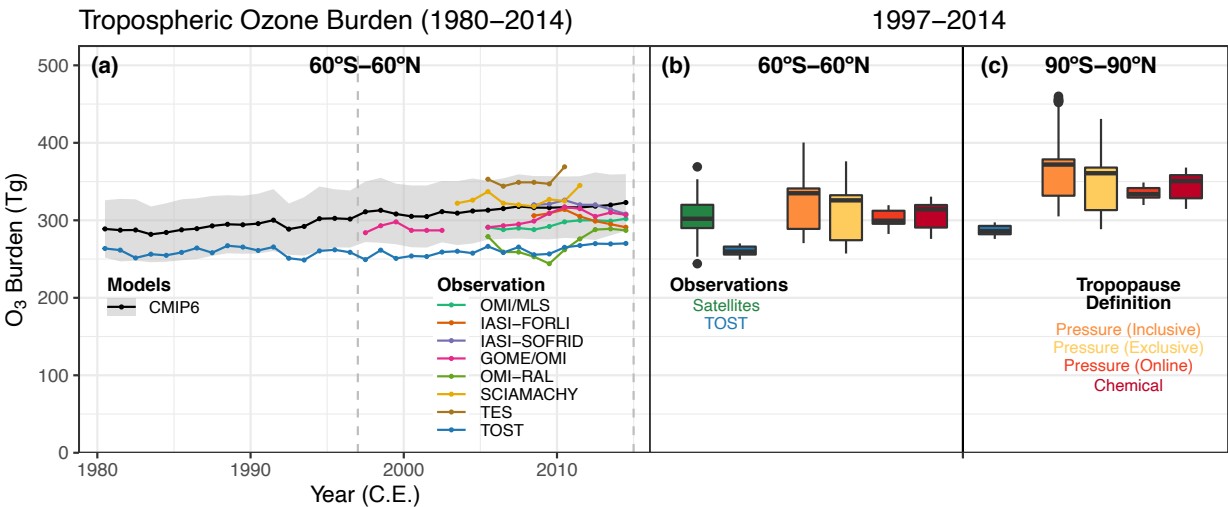

**Figure 7.** Evaluation of the present-day tropospheric ozone burden. (a) Time series of tropospheric ozone burden integrated from 60°S to 60°N for the period 1980 to 2014 (C.E.). The black line shows the CMIP6 ensemble mean using the pressure tropopause excluding the layer which contains the tropopause. The gray shading shows the mean ± one standard deviation of the ensemble inter-model variability for each year. The coloured lines show the annual mean tropospheric burdens reported by seven satellite products aggregated by Gaudel et al. (2018) and the ozonesonde trajectory product (TOST; Liu et al., 2013b, a). (b) Tropospheric ozone burden distribution for 60°S to 60°N for the period 1997 to 2014 C.E., corresponding to the space between the two vertical dashed lines of panel (a). Box-and-whisker plots show the distribution of the various satellite products (green) and TOST (blue), alongside the CMIP6 ensemble using four different tropopause definitions (see main text for details). (c) The same as panel (b), but showing the burden integrated from 90°S to 90°N in the TOST product and models. All units are in Tg $O_3$.

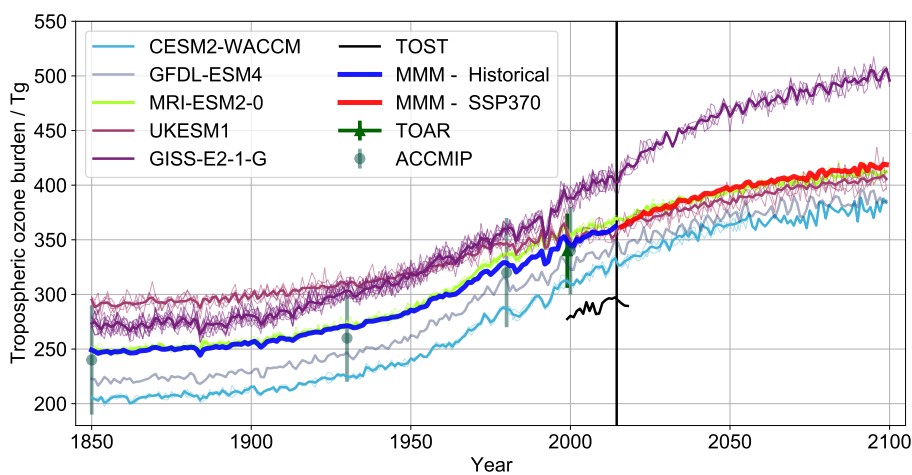

**Figure 8.** Evolution of tropospheric ozone burden integrated from 90°S to 90°N for the period 1850-2100. Models are shown as coloured lines as in the caption. Thick blue line: multi-model mean for CMIP Historical experiment. Red line: multi-model mean for ScenarioMIP ssp370 experiment. TOST burden is show as black line, TOAR multi-model mean as a green triangle and ACCMIP multi-model mean for timeslice experiments as dark green circles.

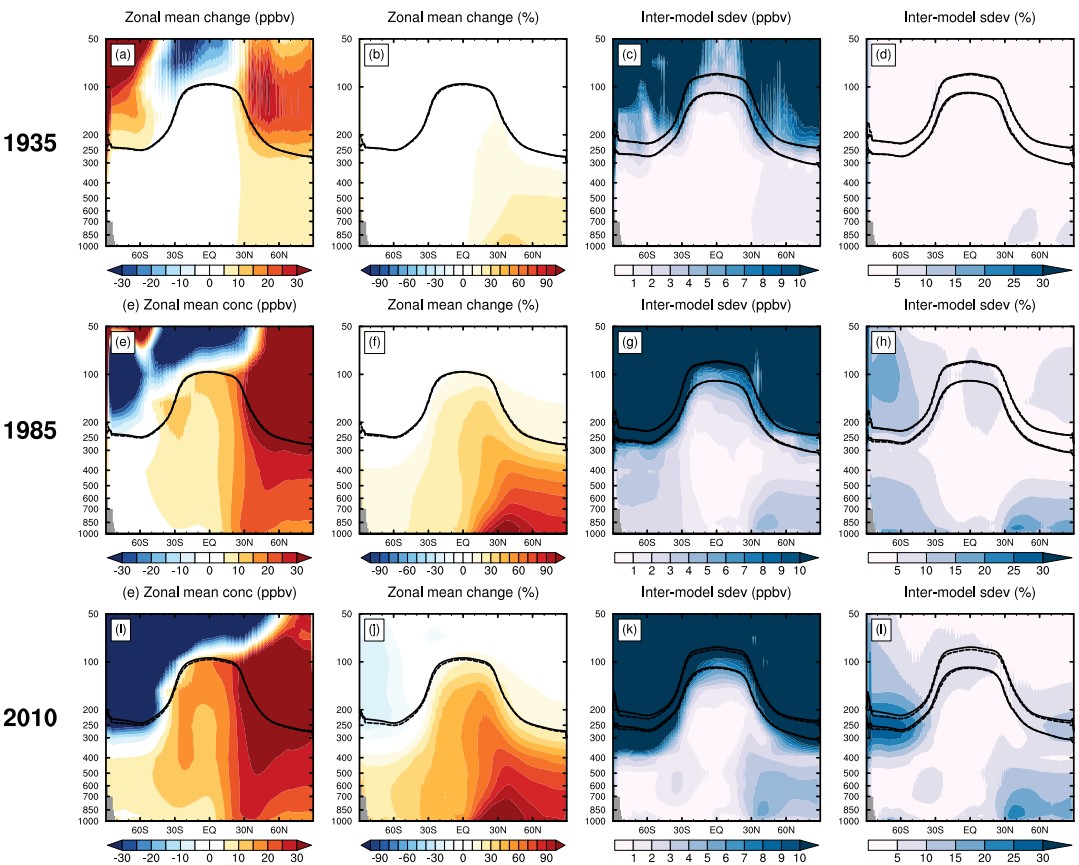

**Figure 9.** Historic change in zonal decadal mean ozone relative to the preindustrial era. Each row shows the change in decadal zonal (i.e. pressure altitude versus latitude) statistics in the CMIP6 historical simulations relative to those of 1850-1859 C.E. From top to bottom: the change at 1930-1939, at 1980-1989, and at 2005-2014 C.E. The left two columns show the absolute and relative change, respectively, in the ozone mixing ratio in nmol mol$^{-1}$ (ppbv) and in percent (%). Both panels show the multi-model decadal mean tropopause pressure for the relevant decade as a solid black line, and from 1850-1859 C.E. as a dashed black line. The second-from-right column shows the absolute inter-model standard deviation in the simulated change in nmol mol$^{-1}$ (ppbv), and the mean $\pm$ one standard deviation in tropopause pressure height in the respective decade (solid line) versus 1850-1859 C.E. (dashed line). The rightmost column is the same as the second-from-right column, but normalized by the multi-model mean in percent (%).

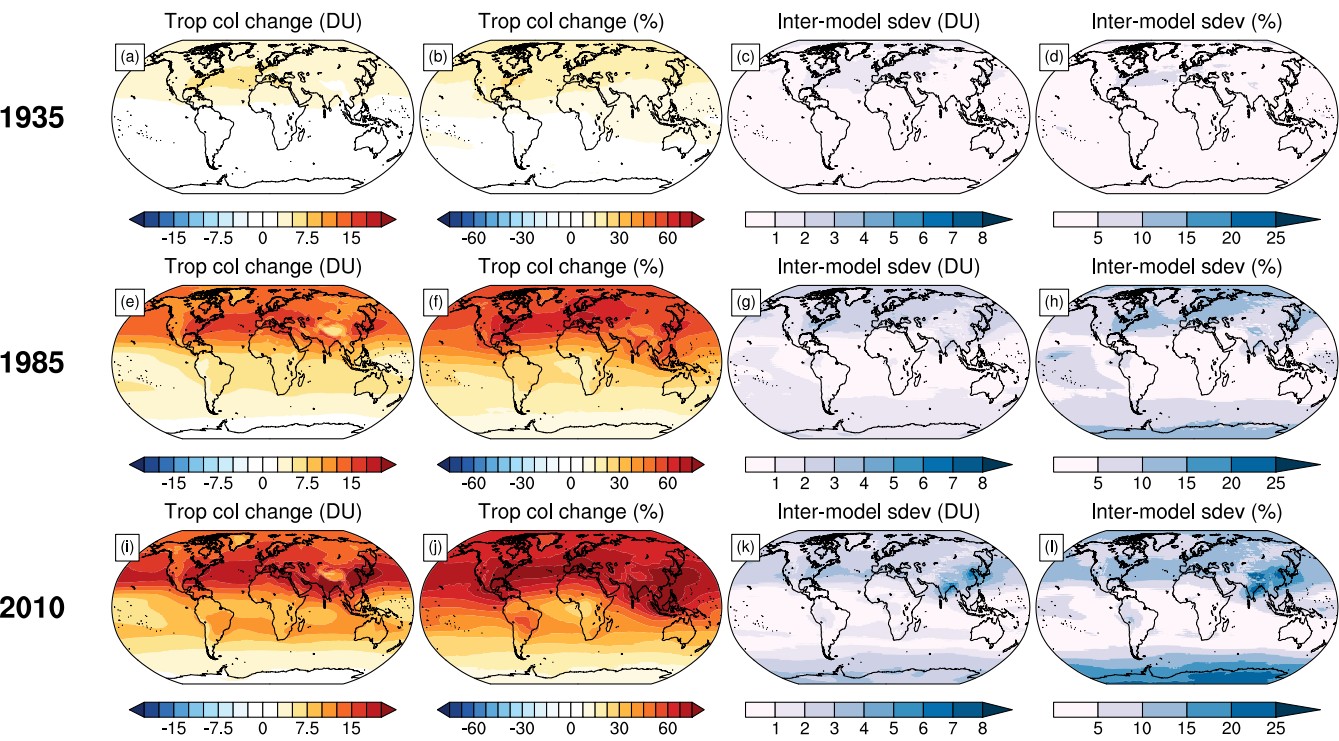

**Figure 10.** Historic change in tropospheric column ozone (TCO) relative to the preindustrial era. The same as Fig. 9, but for changes in TCO in Dobson Units (DU) or percent (%), as appropriate.

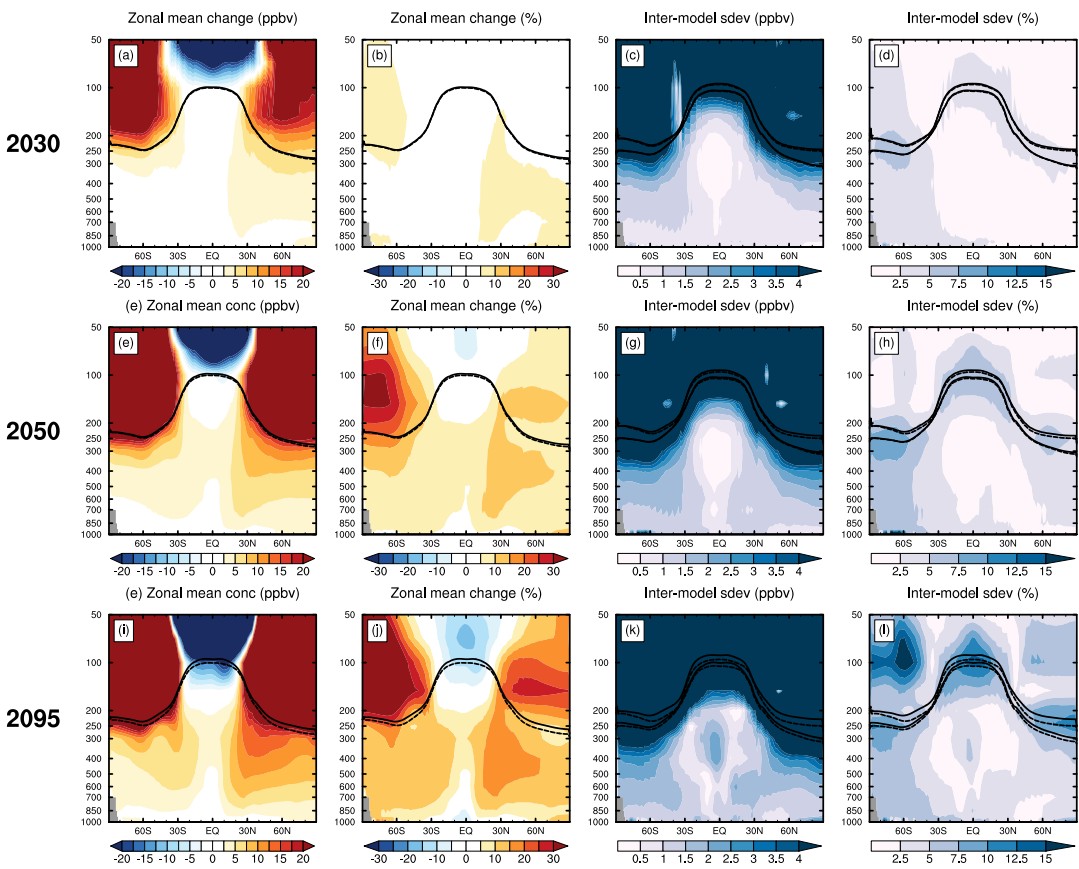

**Figure 11.** Future change in zonal mean ozone relative to the present day. The same as Fig. 9, but showing future decadal statistics in the ssp370 future scenario relative to 2005-2014 C.E. values. From top to bottom: 2025-2034, 2045-2054, and 2090-2099 C.E.

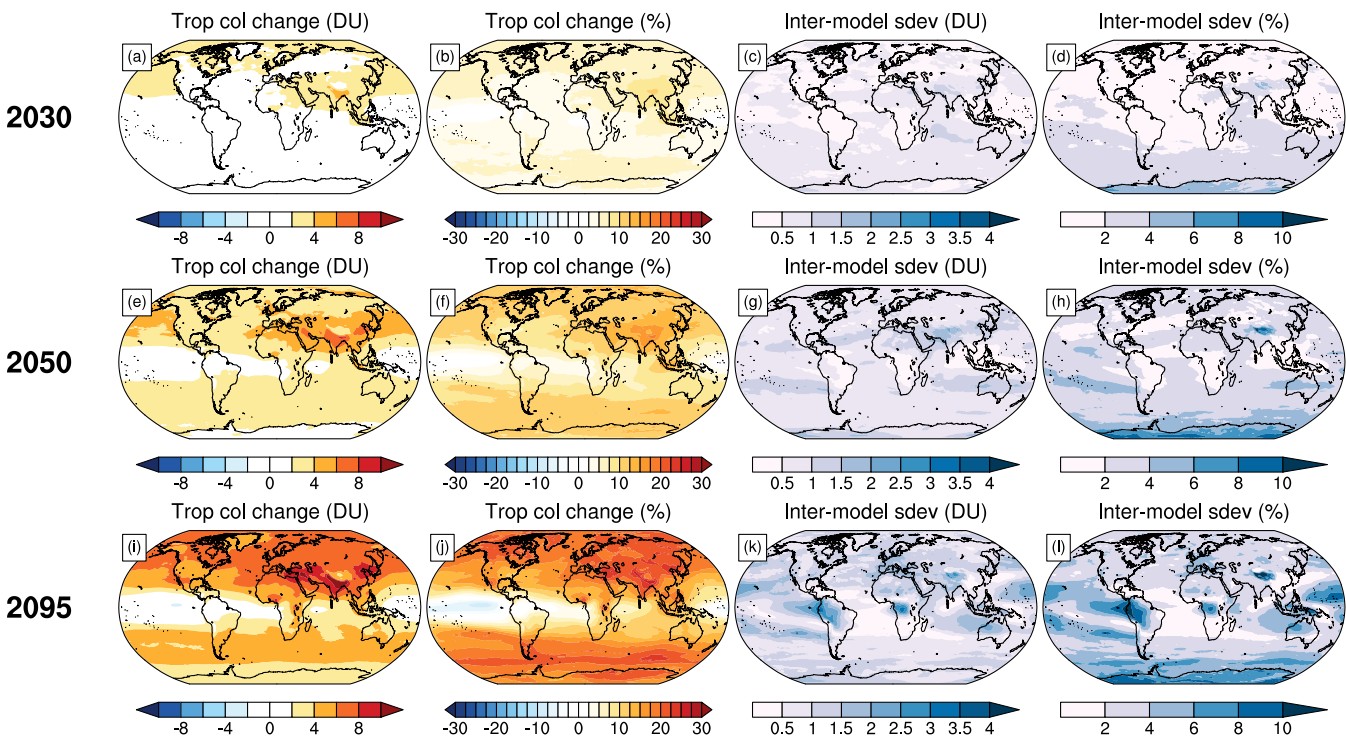

**Figure 12.** Future change in tropospheric column ozone (TCO) relative to the present day. The same as Fig. 11, but for changes in TCO in Dobson Units (DU) or percent (%), as appropriate.

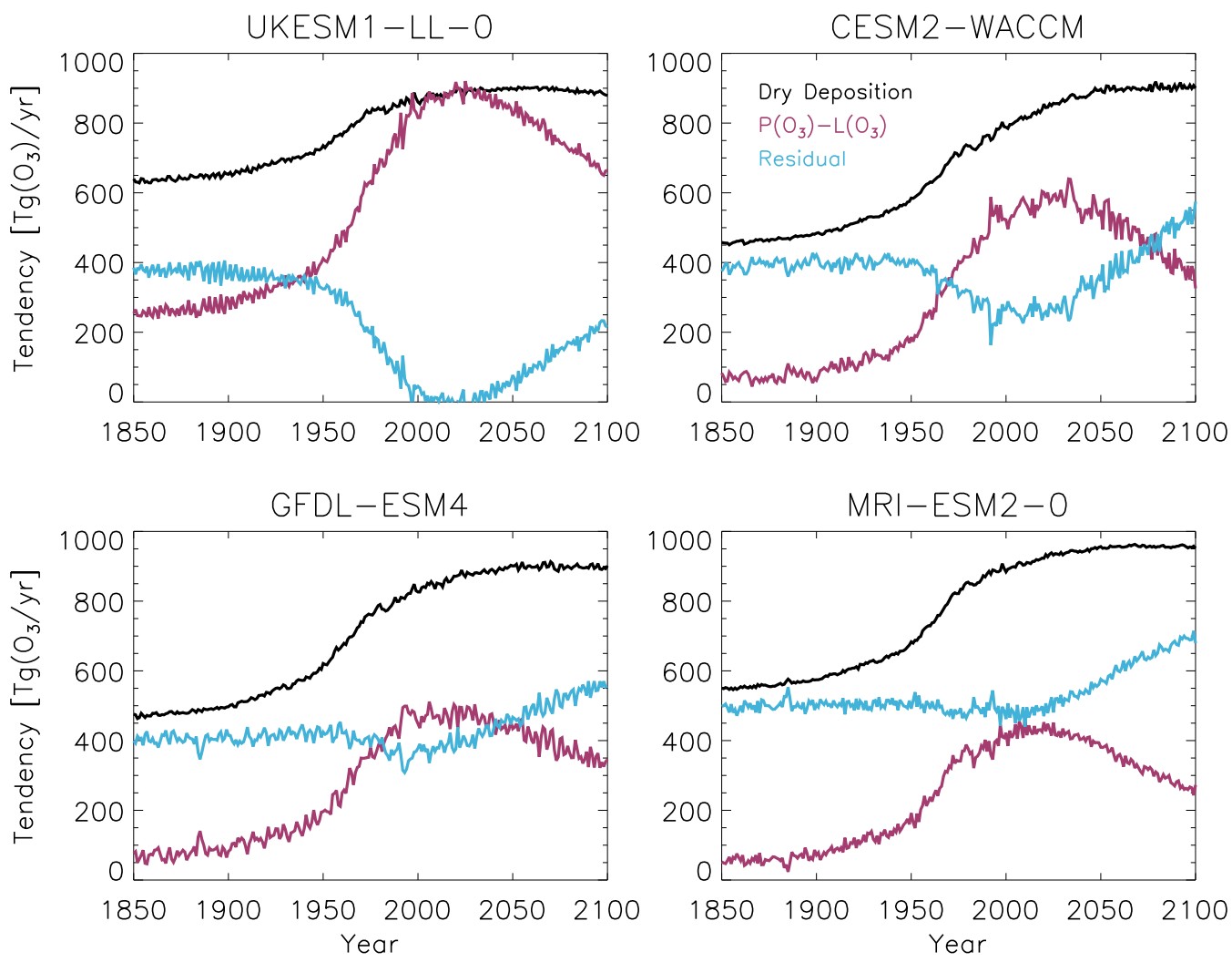

**Figure 13.** Evolution of net chemical production (red line), dry deposition (black line) and residual ozone budget (blue line) over the period 1850-2100 for UKESM1-LL-0, CESM2-WACCM, GFDL-ESM4, and MRI-ESM2-0.

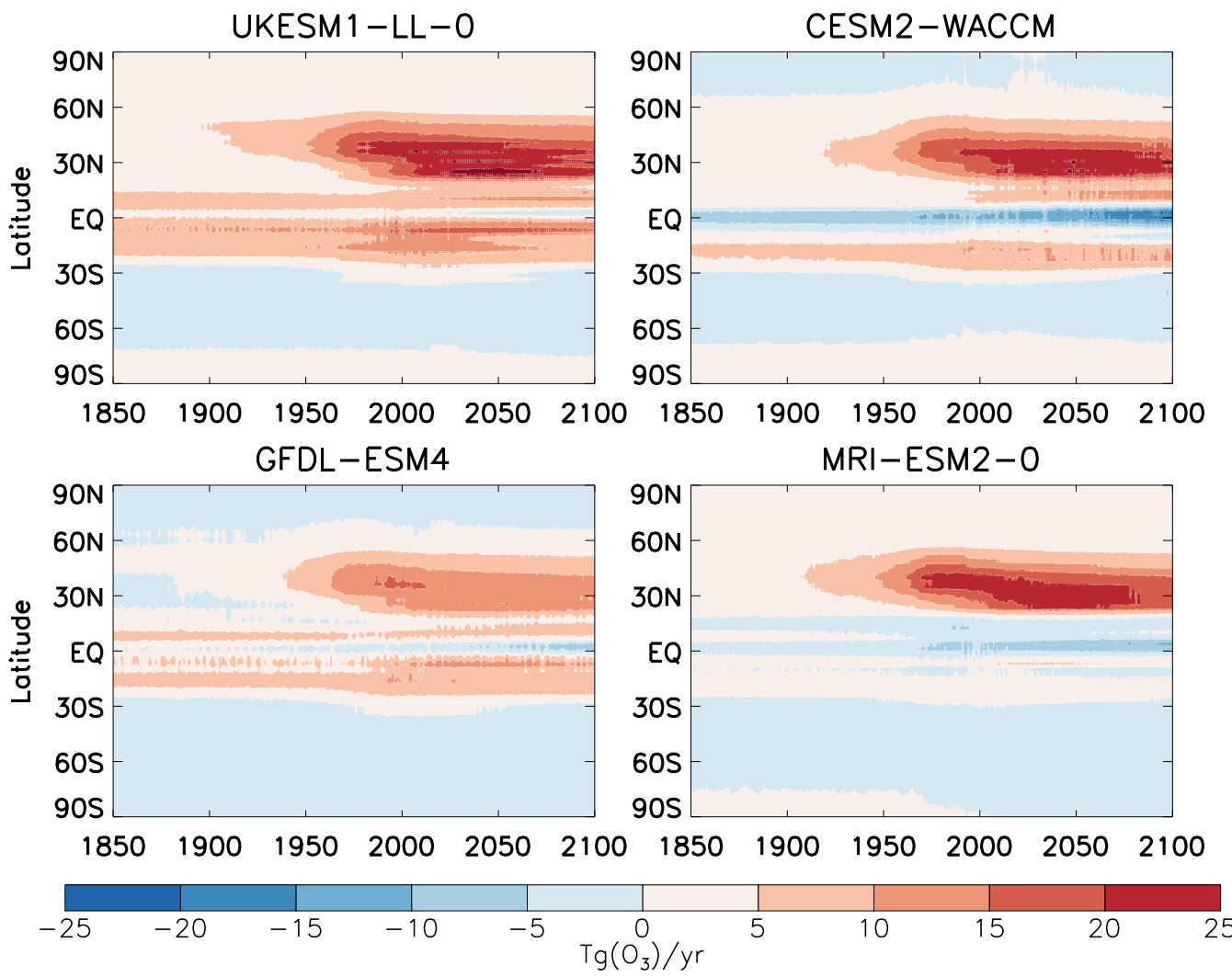

**Figure 14.** Integrated annual net chemical production of tropospheric ozone for UKESM1-LL-0, CESM2-WACCM, GFDL-ESM4, and MRI-ESM2-0. Results are historical (1850-2014) and ssp370 (2015-2100) simulations. Troposphere is masked by the tropopause pressure calculated in each model using the WMO thermal tropopause definition.

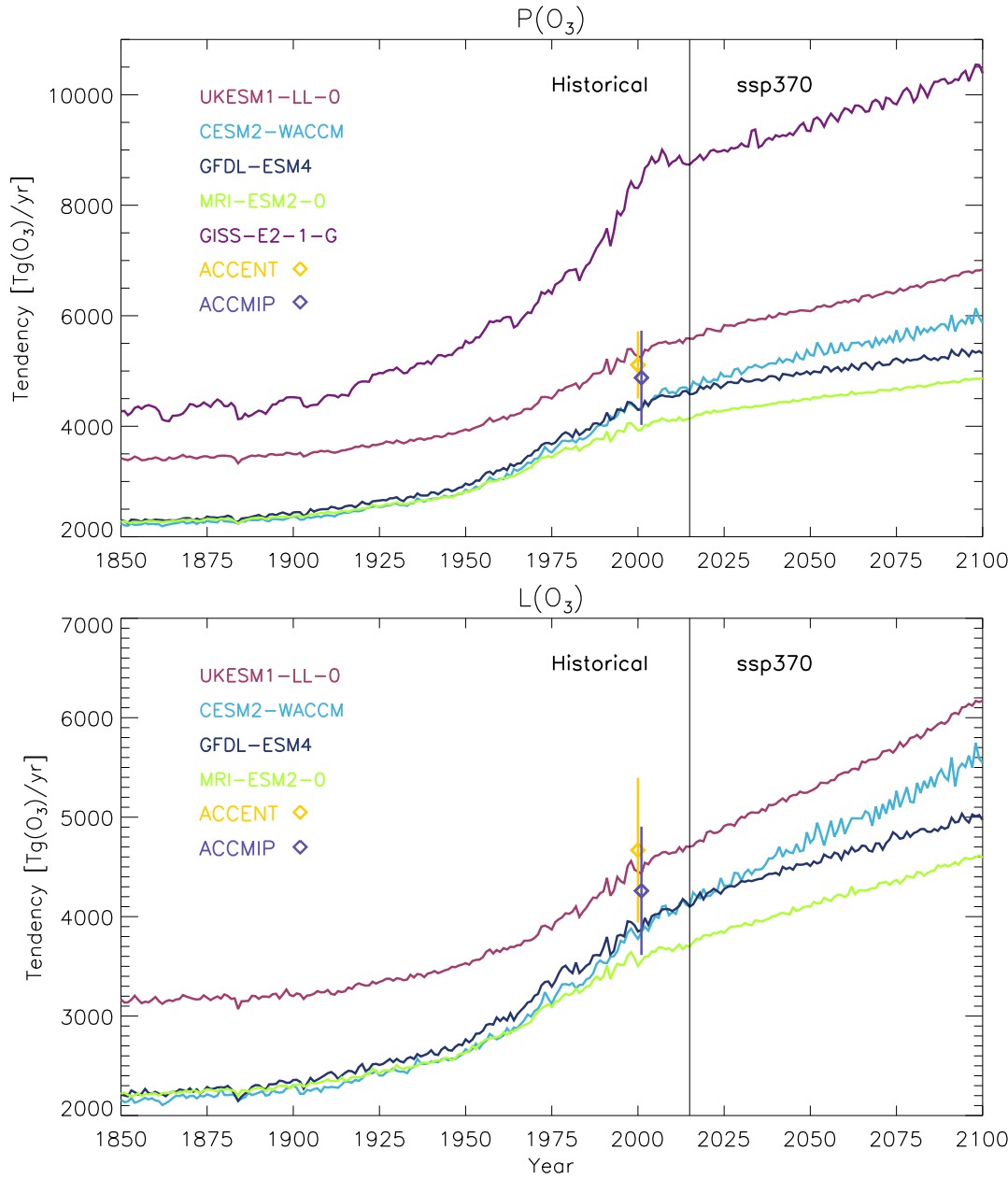

**Figure 15.** Evolution of ozone chemical production (P) and chemical loss (L) terms over the period 1850 - 2100 for the five CMIP6 models (except L from GISS-E2-1-G). ACCENT and ACCMIP production and loss are also displayed for the year 2000, with a slight shift for display purposes.

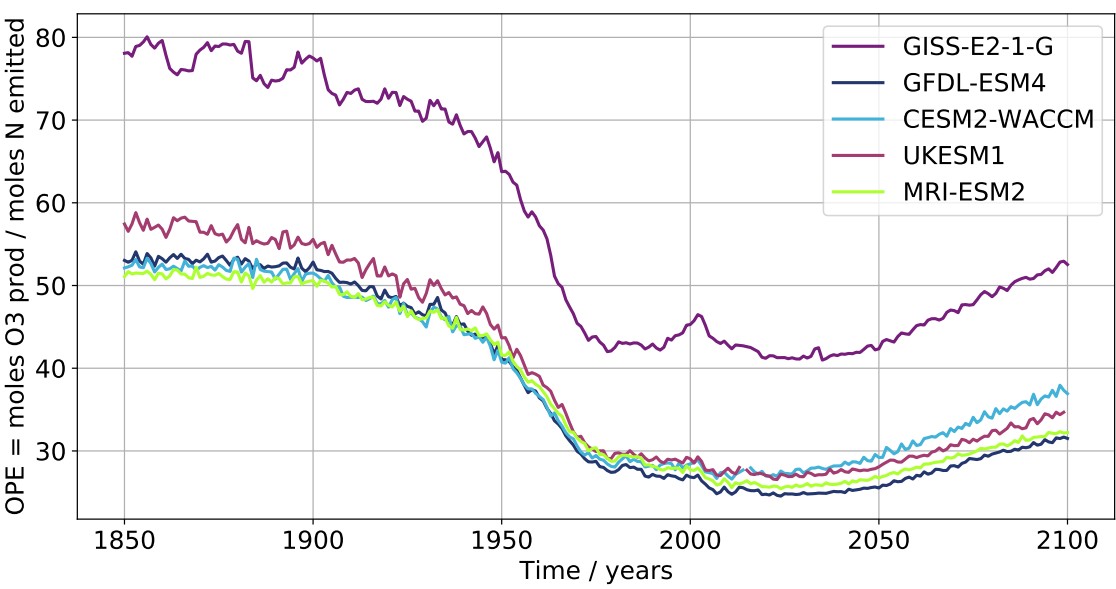

**Figure 16.** Variation in ozone production efficiency (OPE) for the four models. Individual models are shown, as in the figure caption

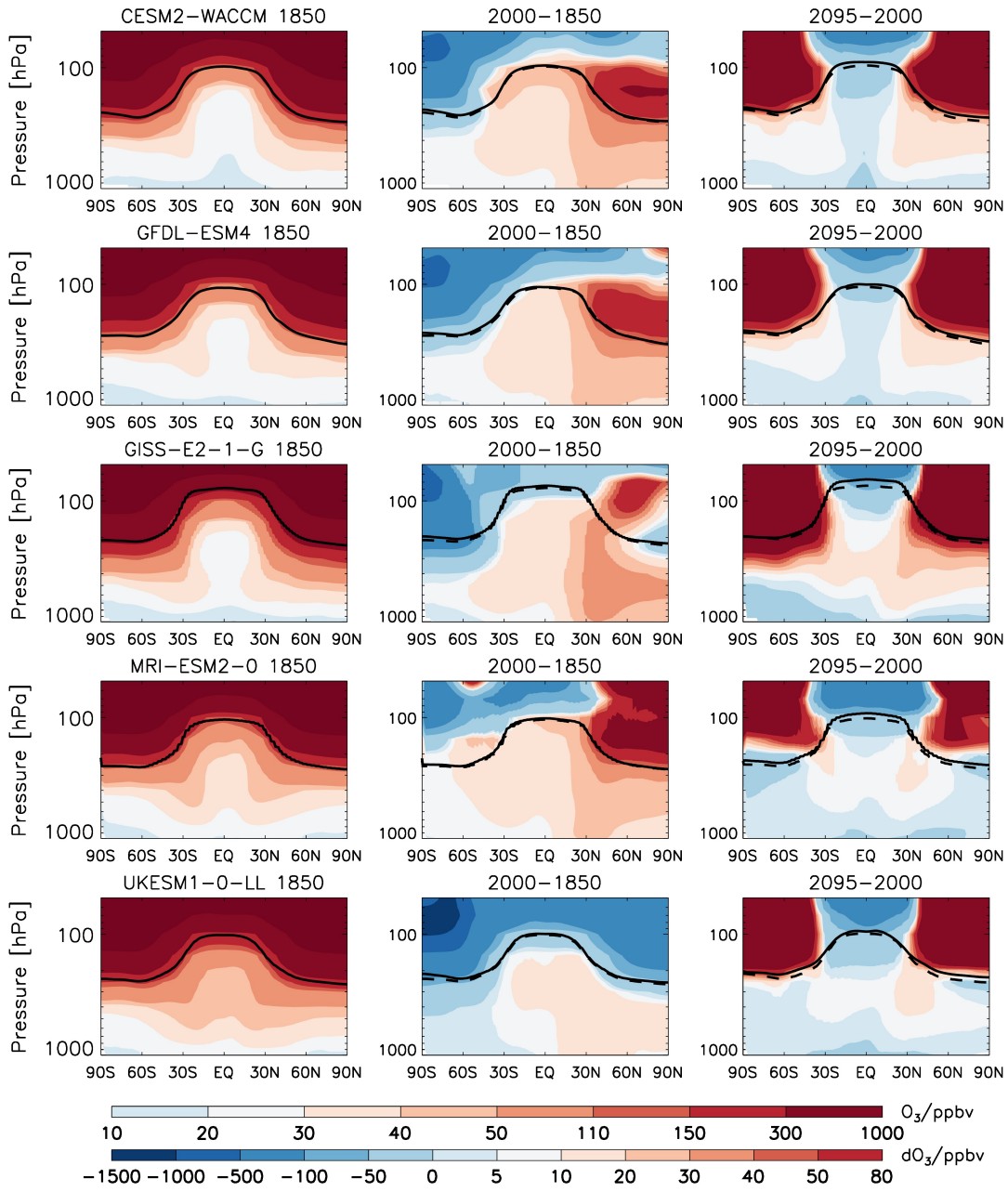

**Figure 17.** Annual and zonal mean ozone distribution in five models over 1850s (averaged over 1850-1859) (left), the difference between 1850 and 2000 (averaged over 1995-2004) (middle), and the difference between 2000 and 2095 (2090-2099) (right). Thick black lines are the tropopause height of each model based on the WMO definition. Dashed black lines are the tropopause for the 1850 period (middle), and for 2000 (right), respectively.