# Peer review of "Tropospheric ozone in CMIP6 Simulations"

_Atmospheric Chemistry and Physics, 2019_

## Referee Comment (RC1) · Mathew Evans (Referee) · 13 Mar 2020

This paper evaluates the simulations of tropospheric ozone from the "preindustrial" (1850), through the present day, to 2100 undertaken as part of the CMIP6 chemistry climate model simulations. This should inform the next IPCC report, and is part of an ongoing multi-decadal project to provide this information for the IPCC reports. The timings for the submission of the paper is mainly driven by the IPCC timescales.

There is utility in publishing this paper. Having a new assessment of both the performance of the current generation of chemistry-climate models and their variability is useful. I would suggest publication after some changes.

There are however some disappointments inherent in this paper which are symptomatic of the CMIP process for tropospheric composition. The comments below are more directed to the wider CMIP community than the authors specifically. 1) The tropospheric

chemistry modelling community appears to be disengaging from this process. Looking at the ACCENT (2006), ACCMIP (2013) and the present study there is a linear decay in the numbers of models taking part in this tropospheric ozone budget aspect. Interpolation would suggest that there will be no models engaging in the process by around 2023. It would be useful for the CMIP community to consider why this is the case, and think about how the outside community is valuing its activities. 2) Papers very similar to this have been being published for the last decades. The authors refer to Young et al. (2013) and Stevenson et al. (2006) as the precursors to this, and there are previous activities which are very similar going back to chapters in the 2001 IPCC report and earlier. It is not obvious that the models' ability to simulate ozone is getting any better over this timescale. One of the conclusions from this paper is the present day mean O3 burden (ACCENT to CMIP6) has only changed by 3% from 15 years of research. 3) It is also of concern that the tools used to analyse these models has not changed in these almost twenty years and the explanations for model differences have similarly not evolved from being a combination of chemistry, emissions, deposition and transport. Perhaps the authors would want to consider whether there needs to be advances in diagnostic techniques before the next model comparison exercise in the conclusions / discussions?

These points are not issues associated with this paper specifically and the authors don't need to reply to these questions but it may be useful for the wider community to think about this.

Specific comments.

Relationship to other CMIP6 papers There are a number of papers submitted to a number of journals based on these CMIP6 simulations. It would be useful to provide some explanation in this paper as to where it is expected to sit in relation to the other papers. Is there a separate paper discussion stratospheric O3? Ozone radiative forcing? OH? CH4 etc. There is some nods to some of these papers but it isn't clear how this is likely to fit in with the other papers. Could the authors provide information about the other

papers currently going through review which touch on this topic (stratospheric ozone, OH etc).

Models 4 models are described in section 2. But I only see 3 models in figures etc. Section 2.1 says that the 'ozone evaluation' uses 4 models but the 'budgets' use 3. But it is unclear therefore why the GISS model doesn't appear in Figure 1,8 and 15. It is unclear which models are being included in which metrics. Could each metric please indicate whether it is calculated from the multi model mean of 3 or 4 models? As discussed earlier this is a small number, especially given previous evaluations. Could the authors give a little bit of context? Are there fewer models engaging in the whole CMIP process, or is it just this tropospheric chemistry aspect aspect? Was the minimum criteria for inclusion solely providing a tropospheric ozone concentration or were there others?

Model description It would be useful to have a table outlining the model configurations. Sections 2.0.1-2.0.4 give differing bits of information about the models and understanding what is the same and what is different between the models is difficult. There are only 3 or 4 models so it shouldn't be too difficult to pull the useful bits of information from the models on things like – resolution, anthropogenic emitted species, lighting emissions scheme, soil nox emissions, biogenic emissions schemes, treatment of aerosols, heterogenous chemistry in a standardized format etc.

Model's representation of tropospheric processes It would be useful for the authors to comment on whether these models represent our best understanding of atmospheric chemistry and, if not, what could the implications of this be. These models are by their very nature fairly conservative in what processes they include and their complexity of representation. But they likely miss some significant processes such as tropospheric halogens, and a complete representation of organic chemistry, heterogenous chemistry etc. It would be useful to have some comments (probably in the discussion) of what this might mean for the conclusions drawn here.

[Figure]

Future and past emissions It would be useful to have a description of the ssp370 emissions – what are the asssumptions about how the world gets to 2100? To those embedded in the IPCC process this might be obvious but to those who are not it is hard to know what this scenario is and what it assumes about the state of the world etc. It would also be useful to have a sense of how the results from these simulations compare to the world predicted by the previous round of model assessments with the RCPs. It would be useful to mark the multi-model mean O3 burdens for 2100 found from the last round of CMIP model experiments on Figure 8 for example. Similarily, how do the preindustrial anthropogenic emissions differ from those previously used in these assessments?

Conclusions The community has been around the cycle of IPCC reported model comparison exercises for tropospheric ozone multiple times now over the last two decades. Figure 8 shows that for the preindustrial to the present day the model prediction (well the multi-model mean) hasn't essentially changed since the ACCMIP evaluation. The explanation for the spread between these models also hasn't really changed. It is some indistinct combination of different emissions, chemistry, deposition and transport in the model. It might be useful to the community for the authors to provide a potential vision of how things should change going forwards. Will the CMIP7 version of this paper look exactly the same as this? If not how should we make advances in the future?

Acronyms There are quite a few acronyms used in the paper. These tend to alienate readers so it would be useful to see if some of them can be removed especially when they are only referred to a few times after being defined. Can the full wording also be used when the definition of the acronym is well away from its usage (on a different page etc). It took me a while to workout what BDC was on page 16.

Specific comments Abstract. Line 10. It's not clear whether the large differences between the models (30%) is referring to the burden or the budget.

Page 16. The OPE calculation is very interesting. This shows a much larger range

than I would have assumed. The authors argue that that it is the differences in the background VOC mixing ratio in the model to explain this. It might not be that simple and they don't really provide any evidence to support this. Differences in the chemistry schemes may play a role here in a number of ways. Choices about the chemical rate constants, mechanistic choices and the speciation of VOCs used could all cause differences They do show that there are differences in the bVOC emissions (Figure 1) but there isn't any other evidence to support their argument about this being due to background VOC mixing ratios (they discuss NMHVOCs in one sentence and then in the next use VOC; is in the other is there a subtly in their argument about CH4 here that I'm not getting). There are also substantial differences in the mean NOx concentrations being calculated which would also influence the OPE. Without some additional evidence the explanations of the model performance appear to be somewhat of a throw away comment.

Tables. Table 1+2 These table are currently without units. It would be useful to include some additional information. The ozone burden would be useful as would the mean lifetime (Burden/(L+DD)). The table seems quite long. Reporting fewer times would not change the story.

Figures.

Figure 1. Could this be expanded to include CH4 concentrations or emissions and the anthropogenic emissions of VOCs, SO2 etc? This would help to put the rest of the paper into context. It would also be useful to know what the models are predicting for OH concentration. I realise this that might be being covered in more detail in another publication but it is hard to understand the impacts of O3 without understanding the influence of OH. If there are other papers covering other areas of the model simulations it would be useful to understand which papers will be covering which activities.

Figure 2. Extra dot before C.E.

Figure 4. The markers are too small to see the colours. I'd suggest that they are just

filled back squares or circles. This seems to be discussed before figure 3?

Figure 5. Can you explain MMS, MMM in the caption text.

Figure 8. Can the models be described in the caption box as well as in the text. It would make it easier to understand.

Figure 14. It would be more useful to know the deposition velocities in the model than the fluxes here. In trying to attribute change it is hard to know whether it's the differences in the O3 concentrations calculated by the model which are causing the differences or the changes in the land surfaces or assumptions about land surface which are causing these differences.

Figure 15. I'm not sure that the units are appropriate here? The units say Tg(O3)/yr but shouldn't the model resolution be taken into account here? The text says that the models are at their native grid so a model at 2x2.5 resolution compared to one at 1x1 would have 5 times as much ozone production in each gridbox which would make it look much redder even if the integrated ozone production was the same? Similarily, will this plot also tend to over emphasise the poles in the budget as it given them equal weight on the plot as the tropics?

Figure 16. Can this be converted into two plots? One of ozone production and one loss? It is a bit busy at the moment.

Figure 17. Can the scale on the plots be changed to reduce the emphasis on the stratosphere and increase the emaphasis on the troposphere? 300 ppbv of O3 is pretty high? Page 16. There are a lot of acronymns here which don't I think make the document transparaent. BDC is defined much earlier in the document making understanding difficult.

Data availability. Can the ESGF be spelt out in more details and a website given?

---

## Referee Comment (RC2) · Anonymous Referee #2 · 31 Mar 2020

This paper provides a current and necessary update to the global tropospheric ozone budget using 3 or 4 state-of-the-art models. The paper will be very useful to the research community, but it first needs a major revision to improve the analysis and discussion in three areas:

1) A major conclusion of Young et al. [2013] is that the projected increase of ozone during the 21st century under RCP8.5 would be almost entirely driven by the large assumed increase in methane. Methane is barely mentioned in this paper, and all focus is placed on BVOCs. It seems unlikely that methane has ceased to be a major factor, and the authors need to discuss the impact of methane on future ozone increases.

2) The paper emphasizes the impact of stratospheric ozone recovery on future ozone increases, but doesn't provide any clear analysis to support this claim. While stratospheric ozone decreases in the mid-latitudes of the southern hemisphere are in the

range of 5-17%, the reduction of stratospheric ozone in the northern hemisphere is quite small, and is less than 5%. Given that the recovery in the Northern Hemisphere will only result in a small increase in the transport of stratospheric ozone into the troposphere, the authors need to provide separate estimates of the impact of ozone recovery on the ozone burden in the Northern and Southern Hemispheres.

3) The model groups did not provide actual flux estimates of the contribution of stratospheric ozone, and instead relied on the outdated and flawed method of estimating the flux based on the residual of the P, L and D terms. Estimates of the stratospheric contribution to the tropospheric ozone budget need to be calculated using a flux-based approach.

I elaborate on these issues in my detailed comments below. Once these issues have been addressed the paper would be acceptable for publication in ACP.

Major Comments:

1) Elaborating on comment #1 above, it would really help if the authors provided a description of ssp370, with a focus on projected methane concentrations. The paper provides no information on this scenario, other than a brief statement in the Conclusions that it is a "middle of the road" pathway. I had to perform a google search, which led me to this paper:

O'Neill, B. C., Tebaldi, C., van Vuuren, D. P., Eyring, V., Friedlingstein, P., Hurtt, G., Knutti, R., Kriegler, E., Lamarque, J.-F., Lowe, J., Meehl, G. A., Moss, R., Riahi, K., and Sanderson, B. M.: The Scenario Model Intercomparison Project (ScenarioMIP) for CMIP6, Geosci. Model Dev., 9, 3461–3482, https://doi.org/10.5194/gmd-9-3461-2016, 2016.

I assume ssp370 must be SSP3-7.0 in O'Neil et al.? According to O'Neil et al. this is a medium to high end scenario with radiative forcing of 7.0 W m-2. This description doesn't really fit with the statement in the Conclusions that this is a "middle of

the road" pathway. As we saw from the ACCMIP results, the factor associated with RCP8.5 that caused ozone to increase over the 21st century was methane. I assume this would also play an important role in the current analysis, but the authors provide no information on the expected methane concentrations; they just say that it increases monotonically. Please provide a description of the expected methane concentrations in ppbv, with a comparison to the current rate of increase, as observed by the NOAA network: https://www.esrl.noaa.gov/gmd/ccgg/trends_ch4/ Please also comment on the relative impact of methane and BVOCs on future ozone levels. On line 403 the authors attribute the ozone increase in the late 21st century to BVOCs. But based on the results of Young et al. [2013] one would assume that methane would be more important. If this is no longer the case, then the authors need to bring BVOCs to the forefront and state very clearly that BVOCs are expected to make a greater contribution to increasing ozone than methane.

2) To provide some background information for my comments in #2 above, here are the latest numbers on observed stratospheric ozone depletion:

Here is the primary link to: "Scientific Assessment of Ozone Depletion: 2018" https://www.esrl.noaa.gov/csd/assessments/ozone/2018/

Here is the link to: "Twenty Questions and Answers About the Ozone Layer: 2018 Update" https://www.esrl.noaa.gov/csd/assessments/ozone/2018/downloads/twentyquestions.pdf On page 46, Figure Q12-1 shows the decrease of ozone in the stratosphere by latitude; the observed reduction in the N. Hemisphere is easily less than 5%.

3) Elaboration on comment #3 above. In the Conclusions (line 554) the authors state: "We find that STE fluxes are similar among the models" However, the authors provide no quantitative support for this statement because they did not actually calculate the flux of ozone from the stratosphere to the troposphere. Even though each of these state-of-the- art models has a fully coupled stratosphere-troposphere circulation, and even though other recent studies have directly calculated the ozone flux, this study

relies on the old, and error-prone, method of simply inferring the flux based on the residual of P, L and D. There are errors associated with P, L and D, and therefore if you rely on these terms to infer the flux from the stratosphere it will reflect all of these errors. An excellent example is the residual term of UKESM1 in Figure 13. The inferred flux from the stratosphere drops to zero in the year 2000, which means that either there is complete ozone depletion in the stratosphere, or there is a complete collapse of the Brewer Dobson circulation. We know that neither of these scenarios is possible, and therefore this inferred flux from the stratosphere is nothing more than errors associated with P, L and D. This study needs to abandon the inference method of estimating STE and use a flux-based method that calculates the net ozone flux across the tropopause, or across the 380 theta isotherm. The 380 isotherm flux method is convenient because any stratospheric ozone that descends from the "overworld" across this layer will eventually enter the troposphere [Holton et al., 1996; Appenzeller et al., 1996]. While there is a delay of several weeks from the time the ozone crosses the 380 isotherm until it crosses the tropopause, it's fine to use this method to calculate an annual average flux. Recent paper that use this method are Jaegle et al., 2017; Olsen et al., 2013; and Yang et al., 2016.

Appenzeller, C., Holton, J. R., & Rosenlof, K. H. (1996). Seasonal variation of mass transport across the tropopause. Journal of Geophysical Research: Atmospheres, 101(D10), 15071-15078.

Holton, J. R., Douglass, A. R., Haynes, P. H., McIntyre, M. E., Rood, R. B., and Pfister, L. (1996) Stratosphere-troposphere exchange, Rev. Geophys., 33, 403–439.

Jaeglé, L. et al (2017), Multiyear composite view of ozone enhancements and stratosphere-to-troposphere transport in dry intrusions of northern hemisphere extratropical cyclones. Journal of Geophysical Research, 122. https://doi.org/10.1002/2017JD027656

Olsen, M. A., Douglass, A. R., & Kaplan, T. B. (2013). Variability of extratropical ozone stratosphere–troposphere exchange using microwave limb sounder observations. Journal of Geophysical Research: Atmospheres, 118, 1090–1099. https://doi.org/10.1029/2012JD018465

Yang, H. et al [2016], Quantifying isentropic stratosphere-troposphere exchange of ozone, J. Geophys. Res. Atmos., 121, 3372–3387; doi:10.1002/2015JD024180

Minor Comments:

Line 20 Need to add uncertainty estimate to ozone RF: 0.4 +/- 0.2 W/mˆ2

Line 34 This statement needs to be reconsidered. Ozone's lifetime is very short and is mostly irrelevant to climate variability on interannual or decadal times scales (e.g. ENSO on a time scale of five years). The impact of climate variability is in relation to shifts in transport pathways and emissions. For example, in strong El Nino years there is increased biomass burning across Indonesia, which boosts ozone production in that region, while ozone decreases on the other side of the Pacific. This seesaw pattern has nothing to do with ozone lifetime and is a direct result of El Nino changing the distribution of ozone precursor emissions. Another way to think about it is in terms of isoprene, which only has a lifetime of a few hours. You can get large fluctuation in isoprene concentrations across the southeast USA just due to the impacts of the seasonal cycle and drought on emissions. You would get similar relative seasonal and interannual fluctuations if isoprene's lifetime was two weeks instead of a few hours.

Line 42 Here the authors state: "Multiple satellite products corroborated by the global ozonesonde network indicate a present-day (2010-2014) tropospheric ozone burden of 338±6 Tg in broad agreement with the current range of model estimates (Gaudel et al., 2018)." Where did the estimate of 338±6 Tg come from? All of the satellite estimates of the tropospheric ozone burden in Gaudel et al. are listed in their Table 5, but this number does not appear in the table. Did the authors take the 3 values (TOST, IASI-FORLI and IASI-SOFRID) from the 2010-2014 column and produce their own range? If so then they need to specify that it relies on just the IASI and TOST (ozonesonde)

products.

Lines 85-104 There is some good discussion here regarding the impact of changes in the BDC on tropospheric ozone. The authors should also consider the following paper that is the first to establish a link between the expanding Hadley circulation and observed changes in tropospheric ozone across southern mid- and high latitudes.

Lu, X., Zhang, L., Zhao, Y., Jacob, D.J., Hu, Y., Hu, L., Gao, M., Liu, X., Petropavlovskikh, I., McClure-Begley, A. and Querel, R., 2019. Surface and tropospheric ozone trends in the Southern Hemisphere since 1990: possible linkages to poleward expansion of the Hadley Circulation. Science Bulletin, 64(6), pp.400-409. http://refhub.elsevier.com/S2095-9273(19)30104-5/h0040

Line 126-127 What is meant by "transported" vs. "non-transported" chemical tracers? Aren't all tracers transported by the model winds?

Line 123 Four models are described, but the basic information on grid resolution and number of vertical layers is only provided for the GISS model. Please provide this information for all four models.

Line 259 Here you should specify that these sites are remote, as there are some urban and rural sites (such as Hohenpeissenberg, Germany, and Whiteface Mountain, New York) that have data since the early or mid-1970s.

Lines 269 and 275 The ultimate source of the surface ozone data is the NOAA Global Modelling Division, and credit should not be given to the person who processed the data (instead mention colleagues who processed data in the acknowledgements). So that the reader can find these data, the following URL needs to be listed in the Data Availability Statement: ftp://aftp.cmdl.noaa.gov/data/ozwv/SurfaceOzone/

Figure 2. This is one of the most important figures in the paper, yet it is difficult to read because the panels are far, far too small. Please expand the figure so that it fills the width of the page.

[Figure]

Figure 4. This comparison should also include the NOAA site of American Samoa, in the marine boundary layer of the South Pacific (-14.2° S, 170.6° W, 42 m) which has continuous data from 1975 to 2015. The data are available here: ftp://aftp.cmdl.noaa.gov/data/ozwv/SurfaceOzone/

Lines 277-278 TOAR-Observations [Tarasick and Galbally et al., 2019] evaluated the historical ozone observations at South Pole (prior to 1974) and only included the 1961-1963 data in their Table 5. The 1964-1966 and 1967-1973 data were not included, presumably because they were not considered to be as reliable. Here it seems that the 1964-1966 and 1967-1973 data were included, and that some type of correction was applied. I don't see mention of these particular correction factors in TOAR-Observations, and they need to be described here.

Line 290 This is the first time that Figure 3 is discussed, but it appears in the text after Figure 4. The numbers of these figures need to match their appearance in the text.

Line 293 data were accessed

Line 293-295 Here the authors state: "A total of 23,392 profiles using Carbon-Iodine (Komhyr, 1969), ECC (Komhyr, 1971), and Brewer-Mast (Brewer and Milford, 1960) sondes from 82 sites world-wide were aggregated over the period 2005-2014." The great majority of the ozone profiles are made using the modern ECC method, rather than the much older carbon-iodide and Brewer Mast methods. TOAR-Observations shows that there are some biases between these methods. Please provide some numbers to indicate the percent of profiles made with the more reliable ECC method.

line 301 There seems to be a word missing after southern hemispheric: "Note that the northern hemispheric overestimate and southern hemispheric seen at the surface..."

Line 311 It's an overstatement to say that satellites provide daily near-global ozone observations. Their orbits don't even provide daily coverage in the tropics, and they can't see through cloud. For global coverage you basically need to build a monthly

composite.

Figure 8 The caption says there is a dark blue line in the figure, but not to my eye. I see light blue (CESM2-WACCM), regular blue (MMM) and gray (is this GFDL-ESM4??).

Line 398 I get an increase of 25%, not 20%, as follows: $100*(350-280)/280 = 25\%$

Line 463 It's not clear which latitude band of the SH you are referring to when you say that ozone destruction reaches a minimum around the year 2000. Are you talking about 40 degrees south? If so, Zhang et al. did not show a shift in emissions from the SH tropics, southward to the SH mid-latitudes. Their Figure 1 in their supplement shows a broad increase of emissions from the equator to 30 or 40 degrees south. In other words, there is not a decrease in the tropics that is balanced by an increase at mid-latitudes (i.e. a shift from one latitude band to another). The latitudinal shift in emissions in Zhang et al. occurred in the NH.

Line 493 What does "shown in 18" mean? Figure 18?

Line 511 Misspelled: UEKSM1

Figure 16 Why are these terms described as fluxes? Flux is the transport of mass across a unit area and will contain units of m-2, as shown by many examples here: https://en.wikipedia.org/wiki/Flux These terms are not fluxes and the y-axis label needs to be corrected. The caption of this figure also contains several typos.

Line 538 It's not clear what is meant by "fluxes". Are you just talking about the deposition flux? The ozone production and loss terms should not be referred to as fluxes.

---

## Author Comment (AC1) · 8 Dec 2020

**Authors' response to reviewer comments on "Tropospheric ozone in CMIP6 Simulations" by Paul T. Griffiths, Lee T. Murray et al.**

This paper evaluates the simulations of tropospheric ozone from the "preindustrial" (1850), through the present day, to 2100 undertaken as part of the CMIP6 chemistry climate model simulations. This should inform the next IPCC report, and is part of an ongoing multi-decadal project to provide this information for the IPCC reports. The timings for the submission of the paper is mainly driven by the IPCC timescales.

There is utility in publishing this paper. Having a new assessment of both the performance of the current generation of chemistry-climate models and their variability is useful. I would suggest publication after some changes.

There are however some disappointments inherent in this paper which are symptomatic of the CMIP process for tropospheric composition. The comments below are more directed to the wider CMIP community than the authors specifically. 1) The tropospheric chemistry modelling community appears to be disengaging from this process. Looking at the ACCENT (2006), ACCMIP (2013) and the present study there is a linear decay in the numbers of models taking part in this tropospheric ozone budget aspect. Interpolation would suggest that there will be no models engaging in the process by around 2023. It would be useful for the CMIP community to consider why this is the case, and think about how the outside community is valuing its activities. 2) Papers very similar to this have been being published for the last decades. The authors refer to Young et al. (2013) and Stevenson et al. (2006) as the precursors to this, and there are previous activities which are very similar going back to chapters in the 2001 IPCC report and earlier. It is not obvious that the models' ability to simulate ozone is getting any better over this timescale. One of the conclusions from this paper is the present day mean O3 burden (ACCENT to CMIP6) has only changed by 3% from 15 years of research. 3) It is also of concern that the tools used to analyse these models has not changed in these almost twenty years and the explanations for model differences have similarly not evolved from being a combination of chemistry, emissions, deposition and transport. Perhaps the authors would want to consider whether there needs to be advances in diagnostic techniques before the next model comparison exercise in the conclusions / discussions?

These points are not issues associated with this paper specifically and the authors don't need to reply to these questions but it may be useful for the wider community to think about this.

**Conclusions**

The community has been around the cycle of IPCC reported model comparison exercises for tropospheric ozone multiple times now over the last two decades. Figure 8 shows that for the preindustrial to the present day the model prediction (well the multi-model mean) hasn't essentially changed since the ACCMIP evaluation. The explanation for the spread between these models also hasn't really changed. It is some indistinct combination of different emissions, chemistry, deposition and transport in the model. It might be useful to the community for the authors to provide a potential vision of how things should change going forwards. Will the CMIP7 version of this paper look exactly the same as this? If not how should we make advances in the future?

**Authors' response:**

We thank the reviewer for the stimulating comments and share his concerns. We would like to respond to the introductory and concluding remarks together.

It may well be that the tropospheric modelling community is less engaged than previously. This may be a result of the increased scope and complexity of the CMIP6 data request, or may reflect the exclusion of chemical transport models from CMIP6 in favour of coupled general circulation/chemistry-climate models. The requirement of the CMIP6 DECK to include idealised CO2 experiments results in an exclusion of models that don't include a coupled carbon cycle.

The centennial scale experiments required for CMIP6 are undoubtedly expensive to perform, and this, coupled with the large number of sub-projects, may have meant that modelling centres have had to be conservative in their participation in order to conserve and prioritize available resources. The timeline of CMIP6 was perhaps also an issue: the expectation that CMIP6 would run over a number of years has meant that the timeline for data availability could perhaps have been clearer. Certainly additional experiments, for example AerChemMIP experiments, that would have contributed to this assessment paper were not available at the time of writing.

We share the reviewer's concern about the progress in the field, and there is little change in the estimates of the present-day ozone burden from CMIP5 to CMIP6. But CMIP6 has made a large step forward in the availability of diagnostic data for calculation of ozone budgets which allow us to understand model diversity through use of consistent production (o3prod variable), destruction (o3loss) and physical removal via dry deposition (dryo3) variables made available through the Earth System Grid Foundation data archive. There is a consistent definition of the tropopause available, which improves model estimates of ozone burden, and is objectively more accurate than the pre-industrial chemopause used for CMIP5. Our ability to diagnose the reasons for inter-model differences is constrained by the lack of required diagnostic output, and should be a focus of future MIPs. Full chemical diagnostic output is not necessarily needed for the whole period, but it is clear from this study that it would be useful to have it for selected time periods, such as 1850, the recent historical period and at the end of the experiments in 2100, where we are particularly interested in the inter-model range of estimates. We have noted this in our revised summary section.

This paper reinforces the point that having the correct diagnostic information is a crucial part of this work. We have added a section to the summary to indicate additional diagnostics, and a potential application of other diagnostic approaches for the calculation of the tropospheric oxidant that may be useful (Bates and Jacob, 2019; Edwards and Evans, 2017). These different approaches may be valuable and should be assessed carefully before CMIP7.

Experimental design is just as important, particularly for understanding the causes of ozone burden change. CMIP6 features a number of useful attribution experiments, mostly in AerChemMIP, to understand the role of e.g., aerosol precursors, VOC and NOx emissions changes (Thornhil et al., 2020). As the reviewer notes, it would be good to complement these with experiments designed to understand the other important aspects of ozone modelling, namely transport and chemistry. On the first point, we should like to see idealised tracer experiments performed, and the inclusion of specific idealised tracers may give a great deal of insight into the model differences driven by dynamics. CCMI employed idealised tracers (Orbe et al., 2018) which have been used to quantify

model differences in circulation, particularly inter-hemispheric transport times and the effects of convection.  E90 tracers (Prather et al., 2011) can be useful for understanding the effects of stratospheric circulation and, while included in CCMI, are not available in CMIP6. For the second point, understanding the different model sensitivities is important.  We note that the sensitivity of different models to emissions has been quantified through the use of multi-model perturbed parameter ensembles in understanding ozone burden (Wild et al., 2020), and that HTAP simulations use a variation in emissions to ascertain sensitivity (Fiore et al., 2009).  Tagged source experiments may also be useful (Butler et al., 2018).  These strategies can be used effectively to quantify the causes of diversity in model response to different perturbations - either in model inputs or in model parameters.  Targeted studies of key processes, such as Hardacre et al. for dry deposition (Hardacre et al., 2015), or as in the ATom project for photochemical ozone models (Prather et al., 2017), would be valuable.  We note that it may also be possible to use the forthcoming AerChemMIP experiments to look at model response to certain emissions changes in a multi-model sense (e.g., Allen et al., submitted).

We can't speculate too much about the CMIP7 paper, but we expect that the goal of CMIP7 will be to calculate the ozone radiative forcing, which will require continued improvement to the estimate of the pre-industrial ozone burden.  Future assessments will need to quantify and understand the roles of biogenic VOC emissions, NOx sources, biomass burning emissions and to provide assessment of changes.  We note that the increasing use of Earth-system models in CMIP6 means that models are increasingly becoming more complex with greater representation of process-level  feedbacks.  An example is dry deposition of ozone which is increasingly treated as an interactive deposition process that  couples to land cover and vegetation.  Biogenic VOC emissions are another, which are increasingly treated in an online sense.  There will be a need to better quantify the source of inter-model variation in such processes, and hence careful choice of diagnostic output is required.

We would urge the community to engage further on these points, particularly in exploiting the CCMI experiments that are scheduled between CMIP deadlines.  It may be advantageous to exploit the CCMI runs further to focus on the process-based understanding of composition and to  quantify inter-model differences in these experiments and so allow CMIP to be retained for the lengthy centennial coupled experiments.

Well designed, well thought out experiments with targeted and novel diagnostics has the potential to gain more traction with the community.  This aspect has been recently discussed extensively by Archibald et al in the TOAR budget paper discussing the role of CMIP DECK style experiments geared towards identifying the roles of changes in chemistry, emissions, deposition in the models leading to differences in modeled tropospheric ozone burden/budget.

**Specific comments.**

Relationship to other CMIP6 papers There are a number of papers submitted to a number of journals based on these CMIP6 simulations. It would be useful to provide some explanation in this paper as to where it is expected to sit in relation to the other papers.

Is there a separate paper discussion stratospheric O3? Ozone radiative forcing? OH? CH4 etc. There is some nods to some of these papers but it isn't clear how this is likely to fit in with the other papers. Could the authors provide information about the other papers currently going through review which touch on this topic (stratospheric ozone, OH etc).

Done. We have added references to available papers published or in discussion.

Models 4 models are described in section 2. But I only see 3 models in figures etc. Section 2.1 says that the 'ozone evaluation' uses 4 models but the 'budgets' use 3. But it is unclear therefore why the GISS model doesn't appear in Figure 1,8 and 15. It is unclear which models are being included in which metrics. Could each metric please indicate whether it is calculated from the multi model mean of 3 or 4 models? As dis- cussed earlier this is a small number, especially given previous evaluations. Could the authors give a little bit of context? Are there fewer models engaging in the whole CMIP process, or is it just this tropospheric chemistry aspect aspect? Was the minimum criteria for inclusion solely providing a tropospheric ozone concentration or were there others?

This has been addressed in the revised manuscript. 5 models are now included and are included in each figure. The reviewer is correct that we used a subset, focusing on those models which include interactive tropospheric ozone and provided data.

Model description It would be useful to have a table outlining the model configurations. Sections 2.0.1-2.0.4 give differing bits of information about the models and understanding what is the same and what is different between the models is difficult. There are only 3 or 4 models so it shouldn't be too difficult to pull the useful bits of information from the models on things like – resolution, anthropogenic emitted species, lighting emissions scheme, soil nox emissions, biogenic emissions schemes, treatment of aerosols, heterogenous chemistry in a standardized format etc.

Done. We now include a table in a supplementary section to this point.

Model's representation of tropospheric processes It would be useful for the authors to comment on whether these models represent our best understanding of atmospheric chemistry and, if not, what could the implications of this be. These models are by their very nature fairly conservative in what processes they include and their complexity of representation. But they likely miss some significant processes such as tropospheric halogens, and a complete representation of organic chemistry, heterogenous chemistry etc. It would be useful to have some comments (probably in the discussion) of what this might mean for the conclusions drawn here.

Done. We will add a section on key uncertainties with regards to missing processes to the summary.

Future and past emissions It would be useful to have a description of the ssp370 emissions – what are the assumptions about how the world gets to 2100? To those embedded in the IPCC process this might be obvious but to those who are not it is hard to know what this scenario is and what it assumes about the state of the world etc.

Done. We have added a section on the SSP370 pathway.

It would also be useful to have a sense of how the results from these simulations compare to the world predicted by the previous round of model assessments with the RCPs. IIt would be useful to mark the multi-model mean O3 burdens for 2100 found from the last round of CMIP model experiments on Figure 8 for example.

This is a difficult point. Firstly, the models used to generate the data for CMIP5 necessarily differ from those used here, which makes comparison difficult. We also use a different tropopause definition, as noted in the text. Secondly, the RCPs used for AR5 do not correspond well to the updated pathways used for AR6 (Figure 2 in Rao et al., 2017) , and neither RCP6 or RCP8.5 correspond well to SSP370 used here. Finally, we have analysed only a single pathway, SSP370, due to data availability at the time of writing, and the range of tropospheric ozone burdens would not necessarily be comparable to the inter-RCP range in CMIP5. We prefer to leave this to a follow-up paper focusing on the different CMIP6 pathways that will be available once more model data come online.

Similarily, how do the preindustrial anthropogenic emissions differ from those previously used in these assessments?

Done. These are given in the Hoesly paper -particularly Figure 2 - which gives a comparison with AR5 - to which we have added a reference, as well as to a recent paper that compares emissions estimates used in CMIP6 (Elguindi et al., 2020).

Acronyms There are quite a few acronyms used in the paper. These tend to alienate readers so it would be useful to see if some of them can be removed especially when they are only referred to a few times after being defined. Can the full wording also be used when the definition of the acronym is well away from its usage (on a different page etc). It took me a while to workout what BDC was on page 16.

We have removed as many acronyms as possible, and checked that they are defined on first use.

Specific comments Abstract. Line 10. It's not clear whether the large differences between the models (30%) is referring to the burden or the budget.

We have clarified this point.

Page 16. The OPE calculation is very interesting. This shows a much larger range than I would have assumed. The authors argue that that it is the differences in the background VOC mixing ratio in the model to explain this. It might not be that simple and they don't really provide any evidence to support this. Differences in the chemistry schemes may play a role here in a number of ways. Choices about the chemical rate constants, mechanistic choices and the speciation of VOCs used could all cause differences They do show that there are differences in the bVOC emissions (Figure 1) but there isn't any other evidence to support their argument about this being due to background VOC mixing ratios (they discuss NMHVOCs in one sentence and then in the next use VOC; is in the other is there a subtly in their argument about CH4 here that I'm not getting). There are also substantial differences in the mean NOx concentrations being calculated which would also influence the OPE. Without some additional evidence the explanations of the model performance appear to be somewhat of a throw away comment.

Done. We have updated this figure with LNOx data that have since come online. The modified figure shows that the intermodel range of OPE values is much smaller when done as a calculation based on

total NOx emissions, which indicates consistent chemistry/response, although the GISS model remains an outlier compared to the other models due to its much higher production term.

**Tables**. Table 1+2 These table are currently without units. It would be useful to include some additional information. The ozone burden would be useful as would the mean lifetime (Burden/(L+DD)). The table seems quite long. Reporting fewer times would not change the story.

Done. Burdens and lifetimes have been added.

**Figures.**

Figure 1. Could this be expanded to include CH4 concentrations or emissions and the anthropogenic emissions of VOCs, SO2 etc? This would help to put the rest of the paper into context.

Done

It would also be useful to know what the models are predicting for OH concentration. I realise this that might be being covered in more detail in another publication but it is hard to understand the impacts of O3 without understanding the influence of OH. If there are other papers covering other areas of the model simulations it would be useful to understand which papers will be covering which activities.

The data the reviewer requests are in Stevenson et al. for the historical period and we have added a reference to this published paper. References to other papers accepted at the time of writing have been added.

Figure 2. Extra dot before C.E.

Done

Figure 4. The markers are too small to see the colours. I'd suggest that they are just filled back squares or circles. This seems to be discussed before figure 3? Figure 5. Can you explain MMS, MMM in the caption text.

Done. Caption text shows the mean and other data.

Figure 8. Can the models be described in the caption box as well as in the text. It would make it easier to understand.

Done, as requested.

Figure 14. It would be more useful to know the deposition velocities in the model than the fluxes here. In trying to attribute change it is hard to know whether it's the differences in the O3 concentrations calculated by the model which are causing the differences or the changes in the land surfaces or assumptions about land surface which are causing these differences.

Done. We have removed this figure.

Figure 15. I'm not sure that the units are appropriate here? The units say Tg(O3)/yr but shouldn't the model resolution be taken into account here? The text says that the models are at their native grid so a model at 2x2.5 resolution compared to one at 1x1 would have 5 times as much ozone production in each gridbox which would make it look much redder even if the integrated ozone production was the same? Similarly, will this plot also tend to over emphasise the poles in the budget as it given them equal weight on the plot as the tropics?

Thanks for the comments. The integrated net ozone production from each model is now on a common grid.

Figure 16. Can this be converted into two plots? One of ozone production and one loss? It is a bit busy at the moment.

Done

Figure 17. Can the scale on the plots be changed to reduce the emphasis on the stratosphere and increase the emaphasis on the troposphere? 300 ppbv of O3 is pretty high? Page 16. There are a lot of acronymns here which don't I think make the document transparaent. BDC is defined much earlier in the document making understanding difficult.

This plot means to demonstrate ozone changes in both the lower stratosphere and the troposphere and we wish to retain this emphasis.

Data availability. Can the ESGF be spelt out in more details and a website given?

Done.

**Bibliography**

Bates, K. H. and Jacob, D. J.: An Expanded Definition of the Odd Oxygen Family for Tropospheric Ozone Budgets: Implications for Ozone Lifetime and Stratospheric Influence, Geophys. Res. Lett., 47(4), doi:10.1029/2019GL084486, 2020.

Butler, T., Lupascu, A., Coates, J. and Zhu, S.: TOAST 1.0: Tropospheric Ozone Attribution of Sources with Tagging for CESM 1.2.2, Geosci. Model Dev., 11(7), 2825–2840, doi:https://doi.org/10.5194/gmd-11-2825-2018, 2018.

Edwards, P. M. and Evans, M. J.: A new diagnostic for tropospheric ozone production, Atmospheric Chem. Phys., 17(22), 13669–13680, doi:https://doi.org/10.5194/acp-17-13669-2017, 2017.

Elguindi, N., Granier, C., Stavrakou, T., Darras, S., Bauwens, M., Cao, H., Chen, C., Gon, H. A. C. D. van der, Dubovik, O., Fu, T. M., Henze, D. K., Jiang, Z., Keita, S., Kuenen, J. J. P., Kurokawa, J., Liousse, C., Miyazaki, K., Müller, J.-F., Qu, Z., Solmon, F. and Zheng, B.: Intercomparison of Magnitudes and Trends in Anthropogenic Surface Emissions From Bottom-Up Inventories, Top-Down Estimates, and Emission Scenarios, Earths Future, 8(8), e2020EF001520, doi:https://doi.org/10.1029/2020EF001520, 2020.

Fiore, A. M., Dentener, F. J., Wild, O., Cuvelier, C., Schultz, M. G., Hess, P., Textor, C., Schulz, M., Doherty, R. M., Horowitz, L. W., MacKenzie, I. A., Sanderson, M. G., Shindell, D. T., Stevenson, D. S., Szopa, S., Van Dingenen, R., Zeng, G., Atherton, C., Bergmann, D., Bey, I., Carmichael, G., Collins, W. J., Duncan, B. N., Faluvegi, G., Folberth, G., Gauss, M., Gong, S., Hauglustaine, D., Holloway, T.,

Isaksen, I. S. A., Jacob, D. J., Jonson, J. E., Kaminski, J. W., Keating, T. J., Lupu, A., Marmer, E., Montanaro, V., Park, R. J., Pitari, G., Pringle, K. J., Pyle, J. A., Schroeder, S., Vivanco, M. G., Wind, P., Wojcik, G., Wu, S. and Zuber, A.: Multimodel estimates of intercontinental source-receptor relationships for ozone pollution, J. Geophys. Res., 114(D4), doi:10.1029/2008JD010816, 2009.

Hardacre, C., Wild, O. and Emberson, L.: An evaluation of ozone dry deposition in global scale chemistry climate models, Atmospheric Chem. Phys., 15(11), 6419–6436, doi:https://doi.org/10.5194/acp-15-6419-2015, 2015.

Orbe, C., Yang, H., Waugh, D. W., Zeng, G., Morgenstern, O., Kinnison, D. E., Lamarque, J.-F., Tilmes, S., Plummer, D. A., Scinocca, J. F., Josse, B., Marecal, V., Jöckel, P., Oman, L. D., Strahan, S. E., Deushi, M., Tanaka, T. Y., Yoshida, K., Akiyoshi, H., Yamashita, Y., Stenke, A., Revell, L., Sukhodolov, T., Rozanov, E., Pitari, G., Visioni, D., Stone, K. A., Schofield, R. and Banerjee, A.: Large-scale tropospheric transport in the Chemistry–Climate Model Initiative (CCMI) simulations, Atmospheric Chem. Phys., 18(10), 7217–7235, doi:https://doi.org/10.5194/acp-18-7217-2018, 2018.

Prather, M. J., Zhu, X., Tang, Q., Hsu, J. and Neu, J. L.: An atmospheric chemist in search of the tropopause, J. Geophys. Res. Atmospheres, 116(D4), doi:https://doi.org/10.1029/2010JD014939, 2011.

Prather, M. J., Zhu, X., Flynn, C. M., Strode, S. A., Rodriguez, J. M., Steenrod, S. D., Liu, J., Lamarque, J.-F.,
Fiore, A. M., Horowitz, L. W., Mao, J., Murray, L. T., Shindell, D. T. and Wofsy, S. C.: Global atmospheric chemistry – which air matters, Atmospheric Chem. Phys., 17(14), 9081–9102, doi:10.5194/acp-17-9081-2017, 2017.

Rao, S., Klimont, Z., Smith, S. J., Van Dingenen, R., Dentener, F., Bouwman, L., Riahi, K., Amann, M., Bodirsky, B. L., van Vuuren, D. P., Aleluia Reis, L., Calvin, K., Drouet, L., Fricko, O., Fujimori, S., Gernaat, D., Havlik, P., Harmsen, M., Hasegawa, T., Heyes, C., Hilaire, J., Luderer, G., Masui, T., Stehfest, E., Strefler, J., van der Sluis, S. and Tavoni, M.: Future air pollution in the Shared Socio-economic Pathways, Glob. Environ. Change, 42, 346–358, doi:10.1016/j.gloenvcha.2016.05.012, 2017.

Wild, O., Voulgarakis, A., O'Connor, F., Lamarque, J.-F., Ryan, E. M. and Lee, L.: Global sensitivity analysis of chemistry–climate model budgets of tropospheric ozone and OH: exploring model diversity, Atmospheric Chem. Phys., 20(7), 4047–4058, doi:https://doi.org/10.5194/acp-20-4047-2020, 2020.

---

## Author Comment (AC2) · 8 Dec 2020

**Authors' response to reviewer comments on - "Tropospheric ozone in CMIP6 Simulations" by Paul T. Griffiths, Lee T. Murray et al.**

This paper provides a current and necessary update to the global tropospheric ozone budget using 3 or 4 state-of-the-art models. The paper will be very useful to the research community, but it first needs a major revision to improve the analysis and discussion in three areas:

1) A major conclusion of Young et al. [2013] is that the projected increase of ozone during the 21st century under RCP8.5 would be almost entirely driven by the large assumed increase in methane. Methane is barely mentioned in this paper, and all focus is placed on BVOCs. It seems unlikely that methane has ceased to be a major factor, and the authors need to discuss the impact of methane on future ozone increases.

2) The paper emphasizes the impact of stratospheric ozone recovery on future ozone increases, but doesn't provide any clear analysis to support this claim. While stratospheric ozone decreases in the mid-latitudes of the southern hemisphere are in the range of 5-17%, the reduction of stratospheric ozone in the northern hemisphere is quite small, and is less than 5%. Given that the recovery in the Northern Hemisphere will only result in a small increase in the transport of stratospheric ozone into the troposphere, the authors need to provide separate estimates of the impact of ozone recovery on the ozone burden in the Northern and Southern Hemispheres.

3) The model groups did not provide actual flux estimates of the contribution of strato-spheric ozone, and instead relied on the outdated and flawed method of estimating the flux based on the residual of the P, L and D terms. Estimates of the stratospheric contribution to the tropospheric ozone budget need to be calculated using a flux-based approach.
I elaborate on these issues in my detailed comments below. Once these issues have been addressed the paper would be acceptable for publication in ACP.

We thank the reviewer for their extensive and valuable comments.

**Major Comments:**

1) Elaborating on comment #1 above, it would really help if the authors provided a description of ssp370, with a focus on projected methane concentrations. The paper provides no information on this scenario, other than a brief statement in the Conclusions that it is a "middle of the road" pathway. I had to perform a google search, which led me to this paper: O'Neill, B. C., Tebaldi, C., van Vuuren, D. P., Eyring, V., Friedlingstein, P., Hurtt, G., Knutti, R., Kriegler, E., Lamarque, J.-F., Lowe, J., Meehl, G. A., Moss, R., Riahi, K., and Sanderson, B. M.: The Scenario Model Intercomparison Project (ScenarioMIP) for CMIP6, Geosci. Model Dev., 9, 3461–3482, https://doi.org/10.5194/gmd-9-3461-2016, 2016.
I assume ssp370 must be SSP3-7.0 in O'Neil et al.? According to O'Neil et al. this is a medium to high end scenario with radiative forcing of 7.0 W m-2. This description doesn't really fit with the statement in the Conclusions that this is a "middle of the road" pathway.

Done. We have added a section describing the SSP-3.70 and corrected the description to 'regional rivalry' and added that the 7.0 Wm-2 is at the high end of the CMIP6 pathways. Reviewer 1 makes a similar point.

> As we saw from the ACCMIP results, the factor associated with RCP8.5 that caused ozone to increase over the 21st century was methane. I assume this would also play an important role in the current analysis, but the authors provide no information on the expected methane concentrations; they just say that it increases monotonically. Please provide a description of the expected methane concentrations in ppbv, with a comparison to the current rate of increase, as observed by the NOAA net- work: https://www.esrl.noaa.gov/gmd/ccgg/trends_ch4/

We have added a panel to Figure 1 showing the CH4 global mean concentrations specified to be used to drive the Historical and SSP370 experiments. The historical CH4 concentrations are from (Meinshausen et al., 2019)

> Please also comment on the relative impact of methane and BVOCs on future ozone levels.

We agree that this is an important point. AerChemMIP contains within it a series of experiments designed to attribute the effect of methane and VOCs for both historical as well as SSP3-7.0 scenario, such as histSST-piCH4 and ssp370SST-lowCH4, which address methane's role specifically, and through the piClim experiments which quantify the impact of BVOC [Collins et al., 2016]. The **relative** impact is difficult to attribute in the context of the CMIP/ScenarioMIP experiments analysed here therefore. For this paper, we excluded AerChemMIP experiments for reasons of multi-model data availability. We anticipate that many follow-on papers will look at this question in more detail and thank the reviewer for the suggestion.

> On line 403 the authors attribute the ozone increase in the late 21st century to BVOCs. But based on the results of Young et al. [2013] one would assume that methane would be more important. If this is no longer the case, then the authors need to bring BVOCs to the forefront and state very clearly that BVOCs are expected to make a greater contribution to increasing ozone than methane.

This is a good point, and we have made the manuscript clearer to emphasise that the increases in the ozone burden in the late 21st century, when NOx and CO stabilise, can be driven by many factors such as methane, climate effects or increase in stratosphere-to-troposphere transport, and added a reference to who discuss the role of methane.

> 2) To provide some background information for my comments in #2 above ["While stratospheric ozone decreases in the mid-latitudes of the southern hemisphere are in the range of 5-17%, the reduction of stratospheric ozone in the northern hemisphere is quite small, and is less than 5%. Given that the recovery in the Northern Hemisphere will only result in a small increase in the transport of stratospheric ozone into the troposphere, the authors need to provide separate estimates of the impact of ozone recovery on the ozone burden in the Northern and Southern Hemispheres."]

We have added a section to the supplementary material showing the data the reviewer requests. It is not possible, without interhemispheric ozone mass flux diagnostic output, to say more about the impact of stratospheric ozone recovery on the hemispheres separately.

3) Elaboration on comment #3 above. In the Conclusions (line 554) the authors state: "We find that STE fluxes are similar among the models" However, the authors provide no quantitative support for this statement because they did not actually calculate the flux of ozone from the stratosphere to the troposphere. Even though each of these state-of-the- art models has a fully coupled stratosphere-troposphere circulation, and even though other recent studies have directly calculated the ozone flux, this study relies on the old, and error-prone, method of simply inferring the flux based on the residual of P, L and D. There are errors associated with P, L and D, and therefore if you rely on these terms to infer the flux from the stratosphere it will reflect all of these errors. An excellent example is the residual term of UKESM1 in Figure 13. The inferred flux from the stratosphere drops to zero in the year 2000, which means that either there is complete ozone depletion in the stratosphere, or there is a complete collapse of the Brewer Dobson circulation. We know that neither of these scenarios is possible, and therefore this inferred flux from the stratosphere is nothing more than errors associated with P, L and D. This study needs to abandon the inference method of estimating STE and use a flux-based method that calculates the net ozone flux across the tropopause, or across the 380 theta isotherm. The 380 isotherm flux method is convenient because any stratospheric ozone that descends from the "overworld" across this layer will eventually enter the troposphere [Holton et al., 1996; Appenzeller et al., 1996]. While there is a delay of several weeks from the time the ozone crosses the 380 isotherm until it crosses the tropopause, it's fine to use this method to calculate an annual average flux. Recent paper that use this method are Jaegle et al., 2017; Olsen et al., 2013; and Yang et al., 2016.

We agree with the reviewer that the residual terms in the ozone budget calculated here, particularly for UKESM1, require clarification. We also agree that a dynamically calculated flux would be a valuable complement, but we note that the CMIP6 data request does not permit the analysis that the reviewer requests, as mass fluxes were not archived.

The use of the residual method is well-established [Stevenson, 2006; Young, 2013; Archibald, 2020], has been shown to agree with dynamical STE calculations [Griffiths et al., 2020] and to be valuable when the global ozone tendency is small [Wu, 2017]. Online mass flux methods are not without their own flaws; e.g., they are prone to double-counting due to rapid changes in tropopause altitude relative to downwelling. Mass flux across a fixed isotherm protects against this, but misses photochemistry in the extratropical lowermost stratosphere. We therefore could not agree wholly with the suggestion to abandon the residual method, nor its association with STE, particularly in its qualitative behaviour over time to emphasise the importance of the stratospheric ozone recovery to the determination of future tropospheric ozone budget and burden. We agree that comparison of the residual method to a dynamical calculation would be valuable, but unfortunately, the diagnostics do not exist.

Nevertheless, to follow up the reviewer's suggestion, we have, where available, sourced directly from the modelling centres the STE fluxes calculated dynamically. The figure below shows ozone transport w.r.t. the WMO tropopause (the CMIP6 PTP variable). It can be seen that the figure shows

good agreement between the magnitude of this term as calculated in the GFDL-ESM4 and UKESM1 models, particularly in the period around 1850, of 300-400 Tg per year, consistent with earlier model estimates, and with the residual in this period, but that GISS-E21-G model, with its consistently higher tropopause, is higher by about a factor of four. In the present day, GFDL-ESM4 and UKESM1 diverge, largely as the result of higher ozone depletion calculated in UKESM1.

The figure, derived from data outside the CMIP6 data request, shows the difficulty in using a meteorological (WMO) tropopause for these flux calculations. This is for two reasons: firstly, the altitude of the WMO tropopause, being a thermal tropopause is difficult to define precisely when there are large regions where the lapse rate is essentially constant, and model vertical resolution tends to be low. In these regions, however, the ozone gradients are high, and so the ozone flux across the tropopause is particularly sensitive to the location of the tropopause.

The use of o3prod, o3loss and dryo3 avoids some of these issues. Given that the bulk of ozone production and loss occurs in the lower troposphere (Archibald and Elshorbany, 2020) the budget terms are less sensitive to the precise location of the tropopause, but there is a significant production of ozone in the UT from lightning NOx. The precise size of this term is sensitive to tropopause definition.

We note that the o3prod and o3loss terms used to calculate the residual include the majority (>90%), but not the total, ozone production and loss channels, and are missing potentially important odd-oxygen loss pathways. The low STE term for UKESM1 may be partly a result of this and the strong ozone depletion simulated in UKESM1 (Keeble et al., 2020; Morgenstern et al., 2020; Skeie et al., 2020) as well as circulation changes (Morgenstern et al., 2020). This is also seen in the supplementary figure, in which the dynamical ozone transport declines to a lower value for UKESM1, compared to GFDL-ESM4, reflecting the greater decrease in lower stratospheric ozone in UKESM1 across the period 1850-2014. As the reviewer notes, changes to circulation may be important, and will be addressed in a subsequent study (Zeng et al., in prep).

We have expanded the discussion of the influence of the stratospheric impact to include the references the reviewer suggests, and to discuss the pros and cons of the two methods, noting the absence of data for dynamical calculations, including a reference to the supplementary figure, and highlighting the potential impact of the missing channels in the o3prod and o3loss terms. We will include the figure below in a supplementary section.

[Figure]

**Minor Comments:**
Line 20 Need to add uncertainty estimate to ozone RF: 0.4 +/- 0.2 W/m^2

DONE: uncertainty added.

Line 34 This statement needs to be reconsidered. Ozone's lifetime is very short and is mostly irrelevant to climate variability on interannual or decadal times scales (e.g. ENSO on a time scale of five years). The impact of climate variability is in relation to shifts in transport pathways and emissions. For example, in strong El Nino years there is increased biomass burning across Indonesia, which boosts ozone production in that region, while ozone decreases on the other side of the Pacific. This seesaw pattern has nothing to do with ozone lifetime and is a direct result of El Nino changing the distribution of ozone precursor emissions. Another way to think about it is in terms of isoprene, which only has a lifetime of a few hours. You can get large fluctuation in isoprene concentrations across the southeast USA just due to the impacts of the seasonal cycle and drought on emissions. You would get similar relative seasonal and interannual fluctuations if isoprene's lifetime was two weeks instead of a few hours.

We have amended the text to say that the ozone lifetime is sufficiently long to be transported over large distances, and that modes of climate variability may affect this transport.

Line 42 Here the authors state: "Multiple satellite products corroborated by the global ozonesonde network indicate a present-day (2010-2014) tropospheric ozone burden of 338±6 Tg in broad agreement with the current range of model estimates (Gaudel et al., 2018)." Where did the estimate of 338±6 Tg come from? All of the satellite estimates of the tropospheric ozone burden in Gaudel et al. are listed in their Table 5, but this number does not appear in the table. Did the authors take the 3 values (TOST, IASI- FORLI and IASI-SOFRID) from the 2010-2014 column and produce their own range? If so then they need to specify that it relies on just the IASI and TOST (ozonesonde) products.

Done.  We have amended the manuscript to make clear that  338 +/-6 Tg refers to the available observations of the whole atmosphere burden derived from TOST and IASI data detailed in Gaudel et al. (2018) as the reviewer notes.  This is necessarily a subset of the burdens derived from other satellite products.

Lines 85-104 There is some good discussion here regarding the impact of changes in the BDC on tropospheric ozone. The authors should also consider the following paper that is the first to establish a link between the expanding Hadley circulation and observed changes in tropospheric ozone across southern mid- and high latitudes.
Lu, X., Zhang, L., Zhao, Y., Jacob, D.J., Hu, Y., Hu, L., Gao, M., Liu, X., Petropavlovskikh, I., McClure-Begley, A. and Querel, R., 2019. Surface and tropo- spheric ozone trends in the Southern Hemisphere since 1990: possible linkages to poleward expansion of the Hadley Circulation. Science Bulletin, 64(6), pp.400-409. http://refhub.elsevier.com/S2095-9273(19)30104-5/h0040

Done. We have added a reference to this article.

Line 126-127 What is meant by "transported" vs. "non-transported" chemical tracers? Aren't all tracers transported by the model winds?

Done. Amended to use the word 'species'

Line 123 Four models are described, but the basic information on grid resolution and number of vertical layers is only provided for the GISS model. Please provide this information for all four models.

Done. We have added a table with these data.

Line 259 Here you should specify that these sites are remote, as there are some urban and rural sites (such as Hohenpeissenberg, Germany, and Whiteface Mountain, New York) that have data since the early or mid-1970s.

Done. We have noted that these are 'remote' stations.

Lines 269 and 275 The ultimate source of the surface ozone data is the NOAA Global Modelling Division, and credit should not be given to the person who processed the data (instead mention colleagues who processed data in the acknowledgements). So that the reader can find these data, the following URL needs to be listed in the Data Availability Statement: ftp://aftp.cmdl.noaa.gov/data/ozwv/SurfaceOzone/

We have added this statement to the Data Availability statement. However, we note that the data used here is not the same as what is directly available from NOAA, as it has been corrected to account for changes in instrumentation over the historical record as described by Tarasick et al. (2019).

Figure 2. This is one of the most important figures in the paper, yet it is difficult to read because the panels are far, far too small. Please expand the figure so that it fills the width of the page.

Done. The PDF of the figure has been scaled to the page width.

Figure 4. This comparison should also include the NOAA site of American Samoa, in the marine boundary layer of the South Pacific (-14.2° S, 170.6° W, 42 m) which has continuous data from 1975 to 2015. The data are available here: ftp://aftp.cmdl.noaa.gov/data/ozwv/SurfaceOzone/

Done. We have added these data to the figure.

Lines 277-278 TOAR-Observations [Tarasick and Galbally et al., 2019] evaluated the historical ozone observations at South Pole (prior to 1974) and only included the 1961- 1963 data in their Table 5. The 1964-1966 and 1967-1973 data were not included, presumably because they were not considered to be as reliable. Here it seems that the 1964-1966 and 1967-1973 data were included, and that some type of correc- tion was applied. I don't see mention of these particular correction factors in TOAR- Observations, and they need to be described here.

Those data were not available at the time TOAR-Observations was written, but they have since been digitized by Sam Oltmans who provided them to Owen Cooper. The corrected Regener chemiluminescent automatic ozone analyser and the ECC analyser both show good agreement with the UV method (see TOAR-Observations, Table 2) and so do not warrant correction.

Line 290 This is the first time that Figure 3 is discussed, but it appears in the text after Figure 4. The numbers of these figures need to match their appearance in the text.

Done

Line 293 data were accessed

Done

Line 293-295 Here the authors state: "A total of 23,392 profiles using Carbon-Iodine (Komhyr, 1969), ECC (Komhyr, 1971), and Brewer-Mast (Brewer and Milford, 1960) sondes from 82 sites world-wide were aggregated over the period 2005-2014." The great majority of the ozone profiles are made using the modern ECC method, rather than the much older carbon-iodide and Brewer Mast methods. TOAR-Observations shows that there are some biases between these methods. Please provide some num- bers to indicate the percent of profiles made with the more reliable ECC method.

Done. This statement has been changed. The reviewer is of course correct that the vast majority of these data are from ECC sondes; in fact all but one site (Hohenpeissenberg), which uses the Brewer-Mast sonde, some data from two Japanese sites, which switched to ECC after 2008, and a small amount of data from two Indian sites, which use a sonde similar to the Brewer-Mast. The Hohenpeissenberg data agree well with ECC data in recent decades, while the Japanese KC data measure a little low (2-6%)[TOAR-Observations]; the behaviour of the Indian sonde in recent decades is not well known.

Iine 301 There seems to be a word missing after southern hemispheric: "Note that the northern hemispheric overestimate and southern hemispheric seen at the surface. . ."

Done. Have removed this sentence.

Line 311 It's an overstatement to say that satellites provide daily near-global ozone observations. Their orbits don't even provide daily coverage in the tropics, and they can't see through cloud. For global coverage you basically need to build a monthly composite.

Done. Modified to 'high frequency'

Figure 8 The caption says there is a dark blue line in the figure, but not to my eye. I see light blue (CESM2-WACCM), regular blue (MMM) and gray (is this GFDL-ESM4??).

Done. We now use CMIP6 standard colours for this plot. (https://github.com/IPCC-WG1/colormaps)

Line 398 I get an increase of 25%, not 20%, as follows: 100*(350-280)/280 = 25%

Amended

Line 463 It's not clear which latitude band of the SH you are referring to when you say that ozone destruction reaches a minimum around the year 2000. Are you talking about 40 degrees south? If so, Zhang et al. did not show a shift in emissions from the SH tropics, southward to the SH mid-latitudes. Their Figure 1 in their supplement shows a broad increase of emissions from the equator to 30 or 40 degrees south. In other words, there is not a decrease in the tropics that is balanced by an increase at mid-latitudes (i.e. a shift from one latitude band to another). The latitudinal shift in emissions in Zhang et al. occurred in the NH.

We agree with the reviewer that there is no shift in emissions from the SH tropics. Rather, we point out that the evolution of net chemical production in the background SH mid-latitudes reaches to its maximum in around 2000. We have rephrased this.

> Line 493 What does "shown in 18" mean? Figure 18?

Done. We apologise if this was not clear.

> Line 511 Misspelled: UEKSM1

Corrected. Thanks again.

> Figure 16 Why are these terms described as fluxes? Flux is the transport of mass across a unit area and will contain units of m-2, as shown by many examples here: https://en.wikipedia.org/wiki/Flux These terms are not fluxes and the y-axis label needs to be corrected. The caption of this figure also contains several typos.
> Line 538 It's not clear what is meant by "fluxes". Are you just talking about the deposition flux? The ozone production and loss terms should not be referred to as fluxes.

Done. We have adopted the term tendency to describe rates of change of quantities. "Fluxes" in Figures 13 and 15 have been replaced with "Tendency".

Bibliography

Archibald, A. T. and Elshorbany, Y. F.: Tropospherc Ozone Budgets: Past, present and future understanding., , doi:https://doi.org/10.1525/elementa.034, 2020.

Keeble, J., Hassler, B., Banerjee, A., Checa-Garcia, R., Chiodo, G., Davis, S., Eyring, V., Griffiths, P. T., Morgenstern, O., Nowack, P., Zeng, G., Zhang, J., Bodeker, G., Cugnet, D., Danabasoglu, G., Deushi, M., Horowitz, L. W., Li, L., Michou, M., Mills, M. J., Nabat, P., Park, S. and Wu, T.: Evaluating stratospheric ozone and water vapor changes in CMIP6 models from 1850–2100, Atmospheric Chem. Phys. Discuss., 1–68, doi:https://doi.org/10.5194/acp-2019-1202, 2020. Morgenstern, O., O'Connor, F. M., Johnson, B. T., Zeng, G., Mulcahy, J. P., Williams, J., Teixeira, J., Michou,

M., Nabat, P., Horowitz, L. W., Naik, V., Sentman, L. T., Deushi, M., Bauer, S. E., Tsigaridis, K., Shindell, D. T. and Kinnison, D. E.: Reappraisal of the Climate Impacts of Ozone-Depleting Substances, Geophys. Res. Lett., 47(20), e2020GL088295, doi:https://doi.org/10.1029/2020GL088295, 2020.

Prather, M. J., Zhu, X., Tang, Q., Hsu, J. and Neu, J. L.: An atmospheric chemist in search of the tropopause, J. Geophys. Res. Atmospheres, 116(D4), doi:https://doi.org/10.1029/2010JD014939, 2011.

Skeie, R. B., Myhre, G., Hodnebrog, Ø., Cameron-Smith, P. J., Deushi, M., Hegglin, M. I., Horowitz, L. W., Kramer, R. J., Michou, M., Mills, M. J., Olivié, D. J. L., Connor, F. M. O., Paynter, D., Samset, B. H., Sellar, A., Shindell, D., Takemura, T., Tilmes, S. and Wu, T.: Historical total ozone radiative forcing derived from CMIP6 simulations, Npj Clim. Atmospheric Sci., 3(1), 1–10, doi:10.1038/s41612-020-00131-0, 2020.

Tarasick, D., Galbally, I. E., Cooper, O. R., Schultz, M. G., Ancellet, G., Leblanc, T., Wallington, T. J., Ziemke, J. R., Liu, X., Steinbacher,M., Staehelin, J., Vigouroux, C., Hannigan, J. W., García, O., Foret, G., Zanis, P., Weatherhead, E., Petropavlovskikh, I., Worden, H. M.,Osman, M., Liu, J. J., Chang, K.-L., Gaudel, A., Lin, M., Granados-Muñoz, M., Thompson, A. M., Oltmans, S. J., Cuesta, J., Dufour, G., Thouret, V., Hassler, B., Trickl, T., and Neu, J. L.: Tropospheric Ozone Assessment Report: Tropospheric ozone from 1877 to 2016,1040observed levels, trends and uncertainties, Elem Sci Anth, 7, 39, https://doi.org/10.1525/elementa.376, 2019.